# The Rise and Evolution of Wind Tower Designs in Egypt and the Middle East

Marian A. Nessim, Aya Elshabshiri [ID], Virginia Bassily, Niriman Soliman, Khaled Tarabieh [ID] and Sherif Goubran *[ID]

Department of Architecture, School of Sciences and Engineering, The American University in Cairo, New Cairo 11835, Egypt; mnesim@nu.edu.eg (M.A.N.); a.elshabshiri@aucegypt.edu (A.E.); vebassily@aucegypt.edu (V.B.); nirimansoli@gmail.com (N.S.); ktarabieh@aucegypt.edu (K.T.)
* Correspondence: sherifg@aucegypt.edu

**Abstract:** Throughout history, vernacular architecture has sought to provide inhabitants with comfort, using local materials and techniques while drawing inspiration from the local culture. This goal has helped natural and passive environmental building techniques to emerge, evolve, and develop. Even though we are increasingly dependent on mechanical ventilation and cooling solutions, passive techniques are in favor due to global climate challenges and the drive toward sustainable construction. One of the most well-known passive cooling techniques is the windcatcher, or wind tower, as it is known in the Middle East (also known as a *malqaf* in Egypt). Windcatchers, which appeared in Egypt during the Pharaonic era, were also present in other vernacular Middle Eastern countries such as Iran and Iraq, and they differed in design and materials. This research aims to extract, analyze, and compare windcatchers throughout historical eras in Egypt and other Middle Eastern countries across three main eras: ancient, medieval, and modern. This study thus provides a timeline for developing these passive cooling systems, demonstrating how they were integrated into architecture over millennia. This study also investigates the design differences in these vernacular models, including their shapes, number of sides, and orientation, and correlates them to climatic and architectural conditions. The results highlight that the vernacular wind towers corresponded to the prevailing wind directions and the ventilation needs of the connected spaces. Furthermore, the findings question the effectiveness and appropriateness of some of the modern incorporations of wind towers, which borrow their design from local precedents.

**Keywords:** wind towers; passive cooling; natural ventilation; Egypt; Middle East; historic analysis

## 1. Introduction

Over the years, human comfort has been a key priority in the process of designing spaces. Before mechanical solutions were available, attempts to achieve this goal were naturally passive. Today, however, even with advancements in mechanical solutions, passive methods remain in favor due to the increasing need for sustainable solutions because of current global environmental concerns. Historically common passive solutions include shading devices, ventilation techniques, and passive cooling and heating techniques. One of the most prevalent cooling and ventilation techniques is the wind tower, also known as a windcatcher. Wind towers are also known as *malqafs* in Egypt, one of the very first countries to implement this feature. Wind towers have become increasingly interesting for researchers due to their potential to create passive cooling and reach target human comfort levels indoors. Wind towers have been in use since the Ancient Egyptian era, primarily used in buildings to facilitate ventilation and indoor cooling. Additionally, windcatchers were used widely during the Egyptian Medieval era and were known as a *malqaf*, literally meaning "catcher".

However, there is a gap in the literature that studied windcatchers in Egypt throughout its history. Most researchers who studied windcatchers in Egypt focused on the medieval

era, especially the Islamic, but none, to our knowledge, thoroughly studied all eras starting from Pharaonic to the modern era going through Coptic and Islamic. In addition, the same gap appeared with research studying windcatchers in the Middle East; it either focused on ancient or modern, but no studies, to our knowledge, went through all eras together. Another new aspect of this current research is analyzing the reason behind the windcatcher form and design, how it differed between regions, and why. Thus, the three primary eras including ancient, medieval, and modern are studied in this research to extract, analyze, and compare windcatchers from different historical periods in Egypt and other Middle Eastern regions. This will contribute to advancing the knowledge of vernacular wind tower designs and bring awareness to their potential as a passive design solution that contributes to sustainability and reduces energy consumption, especially in Egypt. This study offers a timeline for advancing these passive cooling systems, showing how they were incorporated into architecture over centuries. It presents and investigates case studies found in the three eras, explains their main aspects and design differences, including their shapes, number of sides, and orientation, and correlates them to climatic and architectural conditions. This research mainly utilizes the literature and archives to investigate the cases mentioned. Wind roses of each region investigated are then further studied to determine wind towers' correspondence to the prevailing wind directions and the ventilation needs of the connected spaces. The findings also call into question several contemporary windcatcher incorporations that borrow their design from local precedents in terms of efficiency and suitability.

The main purpose of a wind tower is to direct wind toward the desired indoor areas of a building and cool the directed air in the process [1]. Nejat et al. [2] explain, in their research on two-sided wind towers, that wind towers operate on two main forces: wind and buoyancy. The wind force is derived from the difference in wind pressure indoors and outdoors. This feeds into the ventilation function of the wind tower. Buoyancy can be generally understood as the force that pushes objects upward when placed in a fluid [3]. Because hotter air has a lower density than cooler air, it tends to 'float' or rise above the cooler air [4]. Therefore, the force of buoyancy feeds into the cooling function of wind towers [2]. This is because, in buoyancy-driven air shafts, the exhaust vents are placed on the top and connected to a chimney leading them outward, while the inlets are placed lower to allow the cool, dense air to be vented into the space [4]. When the correct number of inlets is provided (such as in multiple-sided wind towers), more wind can be driven into the wind tower to be processed in this way. The number of sides and inlets depends on the prevailing wind directions and wind speed. These operation processes are illustrated in Figure 1.

In short, buoyancy is a force that acts mainly on density [3,4]. This paper's Section 5 will discuss both in more detail.

To provide sufficient ventilation, wind towers are usually built to be tall structures erected on building rooftops [5], enabling them to catch the fast-traveling wind [5] away from the building level below, where the surrounding buildings may obstruct wind movement and slow it down. Moreover, wind towers are directed toward the prevailing wind direction or the direction from which the most significant wind movement occurs [6]. This ensures the highest movement of wind possible, especially on hot summer days when wind movement is typically low. Figure 2 shows a typical Egyptian *Malqaf*.

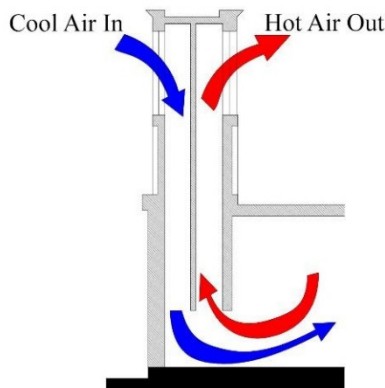

**Figure 1.** A section showing how windcatchers work (modified by the authors; adapted from [7]).

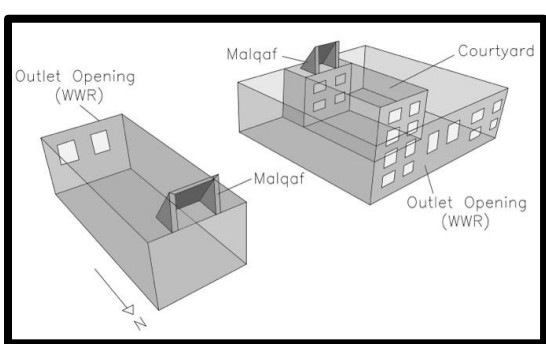

**Figure 2.** The main components of wind towers (modified by the authors; adapted from [8]).

Before the industrial revolution, people living in hot, dry zones deferred to natural cooling methods. Using easily obtainable materials, inhabitants of the Middle East tried alternative methods to ventilate their homes. Windcatchers were widespread in the Middle East, with significant variance in their construction and design. Despite the various structures, they all served the same purpose: to channel favorable prevailing winds into residential areas [9]. Table 1 below shows how different traditional wind tower designs started to appear in different regions [10].

**Table 1.** Traditional wind tower designs and information (modified by authors; adapted from [10]).

| | Egypt | Iran | Persian Gulf | Iraq | Pakistan | Afghanistan |
|---|---|---|---|---|---|---|
| **Climatic zone** | Hot and dry | Hot and dry | Hot and humid | Hot and dry | Hot and humid | Dry and semi-hot |
| **Prevailing wind direction** | Northwest | Northeast | Breeze | Northwest | Southwest | North |
| **Shape of cross-section** | Rectangle | Square/rectangle/hexagon, octagon | Square | Rectangle | Square | Square |
| **Average dimensions (m)** | - | $0.5 \times 0.8$ $0.7 \times 1.1$ | $1 \times 1$ | $0.5 \times 0.15$ $1.20 \times 0.60$ | $1 \times 1$ | $1 \times 1$ |
| **Height (m)** | One story above roof | 3–5 | 3–5 | 1.80–2.10 | 5 and above | 1.5 from roof |
| **Orientation to the prevailing wind direction** | Ordinary | Diagonal | Diagonal | Ordinary | Diagonal | Ordinary |
| **The ceiling of the wind tower** | 30° Slope | 45° Slope | 30° Slope | 45° Slope | 45° Slope | 30° Slope |
| **Ventilated area** | Dining plus one room | Dining room and basement | Dining plus others | Only the basement | All rooms | All rooms |
| **Air flow** | One side | Multi-side | Multi-side | One/two sides | One side | One side |
| **Evaporative cooling** | Sometimes | Sometimes | Never | Sometimes | Never | Never |

In Section 2, this paper starts by reviewing case studies from Egypt showing examples of wind towers from the Pharaonic, Medieval, and Modern eras. In Section 3, this paper presents case studies from the Middle East, which studied historical and modern towers. In Section 4, this paper compares and contrasts the case studies and examines their characteristics vis-à-vis wind rose of the different locations in which they appear. This analysis reveals that the attributes (e.g., number of sides, height, roof shapes, etc.) of historical wind towers have been highly influenced by the weather and wind characteristics of the location, with the modern towers deviating from this trend. This paper then ends with a discussion and conclusions highlighting the key findings, the main limitations of this study, and the areas requiring further investigation and research.

## 2. Case Studies from Egypt (Ancient and Modern)

### 2.1. The Pharaonic Era

Windcatchers have been depicted in Ancient Egyptian paintings, indicating that the concept of the windcatcher traces back to the early Pharaonic era [11]. For example, the Pharaonic house of Neb-Amun was portrayed in a painting on his tomb dating back to the Nineteenth Dynasty (1336–1294 BC). It displays a windcatcher with two openings, one facing windward to receive cool air and the other facing leeward to expel hot air by suction [11,12], as seen in Figure 3a. Furthermore, a papyrus (Figure 3b) from the *Book of the Dead* (1543–1292 BC) references windcatchers' existence during the Pharaonic era. The elevation at the right end of the drawing shows two similarly aligned triangles, which are presumably windcatchers [13].

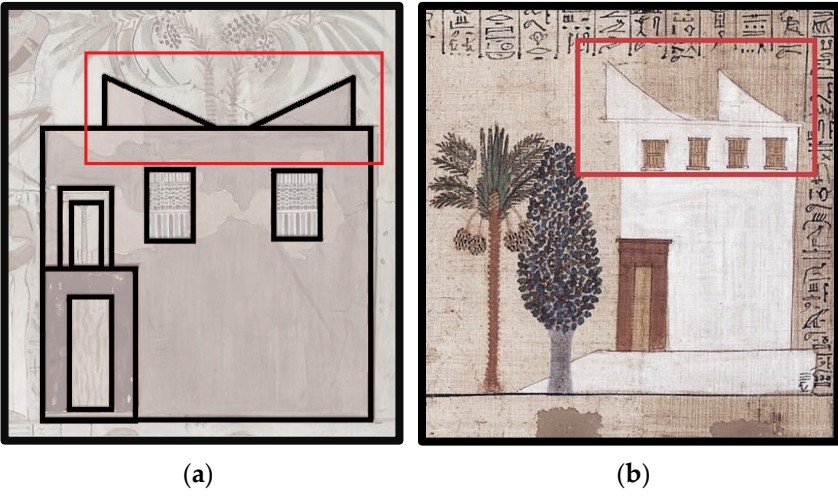

(**a**)          (**b**)

**Figure 3.** (**a**) Windcatchers of the Pharaonic house of Neb-Amun, recreated by authors based on [14], (**b**) *Book of the Dead* showing a building with a wind tower at the lower right corner shown in the red square (about 1336–1294 BC) (modified by the authors, based on [13]).

A papyrus displaying the floor plan of a house constructed in the second century BC was one of the papyri found "in a former dump near Oxyrhynchus in Egypt" (modern el-Bahnasa) (Figure 4). By "extracting their volumes from the redrawing of the plan and the average height typical of the interiors' volume, the speed was 1.5–2 m/s, assuming that the chilly air inside was descending at that rate. This indicates that it takes about 4–5 s to completely refill the air. It is important to note that the presence of an atrium significantly increases internal comfort and improves efficiency [13].

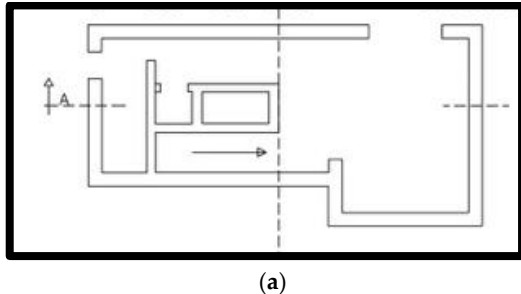 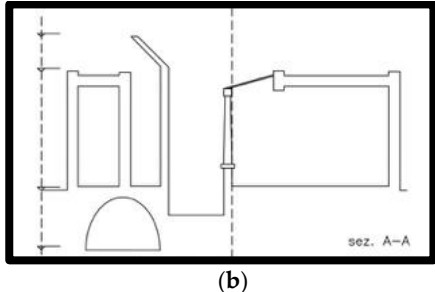

| (a) | (b) |

**Figure 4.** (**a**) Ground plan and (**b**) section of a house found on a papyrus that dates back to the second century BC and is currently located in the Museum of Art and Archeology, Oxford (modified by the authors; adapted from [13]).

### 2.2. Medieval and Ottoman Eras

Windcatchers were prominent in Cairo during the Fatimid and Mamluk eras, from the 10th century to the 19th century. However, according to King, the origins of Medieval Cairo's windcatchers are unknown. They could have been introduced during the time of the new city's establishment in 969 AD or shortly afterward [12]. According to Hassan Fathy's book, *Architecture for the Poor* (1973), the old houses in Cairo relied on windcatchers for ventilation in the principal halls (*qa'as*) that caught the wind at a high elevation "where it is strong and clean". As shown in Figure 5, the hot air escaped through the high central part of the *durqa'a* (a small, covered court) [15].

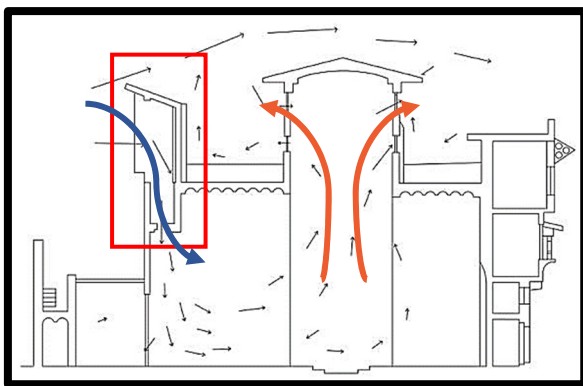

**Figure 5.** Arrows showing the air movement in Qã'a of Muhib Al Din Ash-Shãf'i Al-Muwaqqi with a red box highlighting the location of the wind catcher (modified by the authors; adapted from [16]).

The primary typologies of buildings with windcatchers were mosques, *madrasas* (religious schools), and mausolea. However, relatively few windcatchers were observed after several were restored around 1900 AD [12]. According to David King [12], the medieval windcatcher in Cairo is the most overlooked historical Islamic architectural feature in the mainstream literature, but it was significantly used in Cairo (known then as Fustat) from the 10th to 19th centuries. During this time, most of the homes in Cairo were built with windcatchers on their roofs to drive the cool northern wind down to refresh the living quarters below [12]. Pages 125–128 of King's book show photographs dating back to the 19th century, depicting the presence of windcatchers all over the city [17].

The medieval windcatchers in Cairo were designed to capture the cool northern wind, as can be concluded from their unique shape. According to Williams (2008), a medieval windcatcher resembled the head of a stairway, the shape of which dates back to the Pharaonic era [18]. This distinguishing feature emerged beyond a flat roof, usually at a 30-degree angle. A simple medieval windcatcher consisted of a lightweight wooden rectangular awning that covered an opening in the ceiling of the room below. It was frequently connected with a vertical duct made of masonry or a light structure that directed the flow deep into the building's lowest floors. The opening into the room was either

a horizontal opening through the ceiling or a vertical opening in one of the walls. The windcatcher successively supplied rooms on different levels, whereby the first openings of the column or duct were through vertical openings in the wall, and the last was opened through a horizontal opening in the ceiling [19,20]. Windcatchers could also be built into the windward façades of a building, which differ very little from simple windows [19].

Medieval writings do not mention the materials used to make Cairo windcatchers. The oldest surviving example of a medieval windcatcher (shown in Section 2.2.7, is on the roof of the Qã'a of Muhibb al-Din), which indicates that the windcatchers could have been made of stone, thereby enabling them to survive for centuries. However, according to 19th-century paintings, they were primarily made of wood or reed and were plastered on both sides and hence, had a short lifespan [10]. Nonetheless, there are still intact grilles on the ceilings of historical buildings, indicating that windcatchers were not only present but prevalent [12]. Furthermore, to filter dust from the air, windcatchers were protected with bay wood [21].

Moreover, Olivier Jaubert's observations [19] showed that using wooden shutters on the upper aperture at the roof level controlled airflow and the shutting of the windcatcher [12]. Jaubert further discussed systems for controlling and closing the windcatcher during the medieval period [19]. The devices could be a door leaf on an opening fitted in the wall, horizontal shutters placed on the opening in the ceiling or halfway up the duct, or windows and shutters made of wood and directly applied to the inlet of the windcatcher, which delimited the northern opening of the windcatcher [19].

The windcatchers ranged in size from small structures to immense structures taking up the entire top floor of the building on which they were installed (as seen in the House of Alfi Bey in Section 2.2.9 and Musāfirkhāne Palace illustrated in Section 2.2.10) [12]. According to Ibn Yunus, a medieval astronomer, the rectangular base of the windcatcher is suggested to have dimensions of 10:5 1/2. For example, the windcatcher on the Musāfirkhāne is approximately twice as wide as deep, as shown in Figure 6a. On the other hand, those recommended by Najm al-Dīn al-Miṣrī, another medieval astronomer, were approximately 4:1, as shown in Figure 6b [12]. The following sub-sections of this manuscript present several examples of Medieval Egyptian windcatchers in various building typologies.

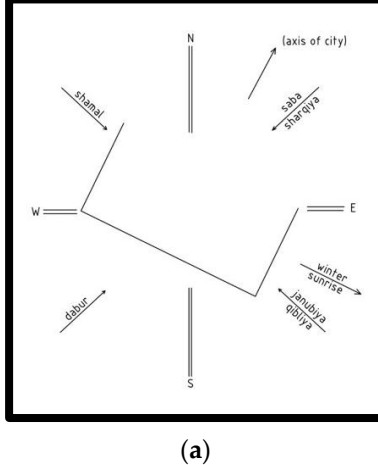 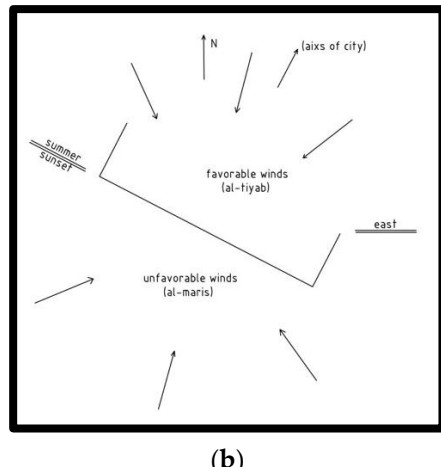

| (a) | (b) |

**Figure 6.** (**a**) Plan showing the range of wind directions in Egypt, according to the medieval 14th-century encyclopedist al-Qalqashandī, in relation to the orientation and shape of windcatchers defined by Ibn Yūnus four centuries earlier. (**b**) Plan showing the range of favorable wind directions in Egypt according to Najm al-Dīn al-Miṣrī (modified by the authors; adapted from [17]).

2.2.1. Christian Hermitages in the Desert of Esna (Upper Egypt), Fifth–Sixth Centuries

Hermitages consist of a courtyard and peripheral rooms dug into the desert ground. Previous surveys carried out on hermitages do not provide evidence for the use of windcatchers. However, ventilation methods were found in these hermitages. Ventilation ducts

were arranged to establish the necessary cross ventilation since the entire hermitage was dug into the ground, and the rooms had only one façade in the courtyard on the opposite side of which there was a ventilation duct. The reserves usually had a ventilation chimney (Figure 7). Doors open into the courtyard and were exposed to the wind.

Similarly, horizontal ducts were sometimes circular and flared like a funnel open to the outside of the courtyard (Figure 8). Reliable archaeological traces confirm that this duct was equipped with a wooden shutter closing system. The direction of the wind determined the relative position of the different parts of the hermitage [19,22].

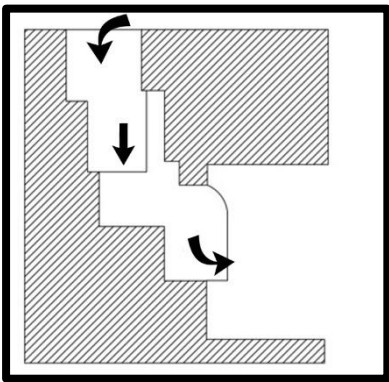

**Figure 7.** Illustrative sections of a reserve ventilation chimney, the arrows signify the air flow (modified by the authors; adapted from [22]).

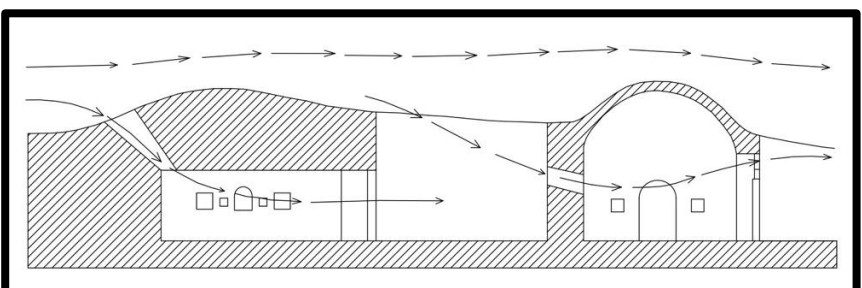

**Figure 8.** Airflow in a hermitage in the desert of Esna, which was built in the ground, the arrows signify the air flow (modified by the authors based on [22]).

2.2.2. Christian Hermitage (Building No. 45), the Kellia, Fifth–Seventh Centuries

The Kellia hermitage included a non-central courtyard and half-excavated rooms. The rooms were generally attached to the northwest corner and along the north wall on the long side, exposed to the prevailing wind. "Amphora necks" integrated into the masonry allowed air to circulate from one room to another. In the roof, two or more holes were present on the domes, depending on the room size. On the north side, the hole was surmounted by a small conch-shaped edicule designed to catch the wind, and on the leeward side, a simple or round opening allowed the air to escape (Figure 9) [19].

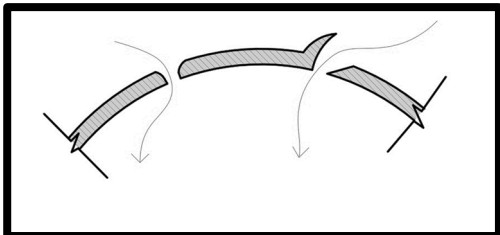

**Figure 9.** Christian hermitage (building no. 45), the Kellia, the arrows signify the air flow (modified by the authors; adapted from [19]).

### 2.2.3. Virgin Mary Hanging Church (Seventh Century)

Figure 10 shows two existing windcatchers in the Virgin Mary Hanging Church. The windcatchers are located above the *narthex* or lobby, facing northeast [23].

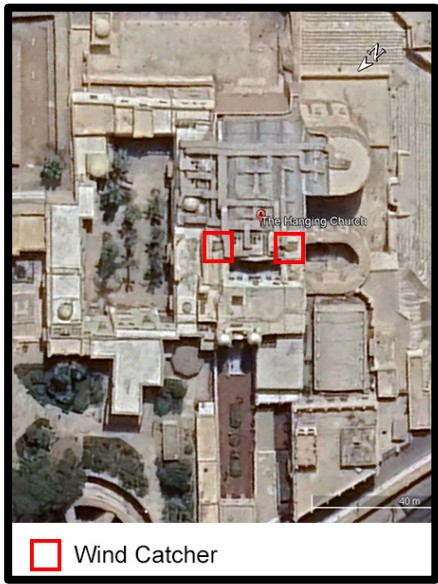

**Figure 10.** Virgin Mary Hanging Church, top view (taken from Google Earth based on [23]).

### 2.2.4. Virgin Mary Al-Damshareya Church (Eighth Century)

Windcatchers can be seen in Figure 11. They are located above the *narthex* or lobby, facing northwest and above the south chapel [23].

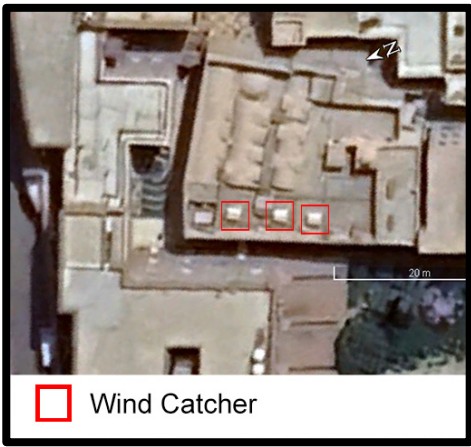

**Figure 11.** Virgin Mary Al-Damshareya Church, top view (taken from Google Earth based on [23]).

### 2.2.5. Mosque of Al-Ṣāliḥ Ṭalā'i' (1160 AD)

In the Mosque of Al-Ṣāliḥ Ṭalā'i', immediately behind the Imam's pulpit is a rectangular opening 71 cm wide and 1.82 m high that is fitted with a grille. The opening opens into a rectangular 0.5 m² vertical shaft (Figure 12). The shaft ascends through the thickness of the wall until it reaches the roof to an awning facing north [18,19,24,25]. Similarly, the Madrasa of al-Nāṣir Muḥammad (1295–1304) has an opening to the right of the mihrab of the Madrasa and a vertical shaft embedded in the masonry of the wall, 1.72 m deep, which are what remains of the windcatcher installed in the main north-western *Iwan* [19,24].

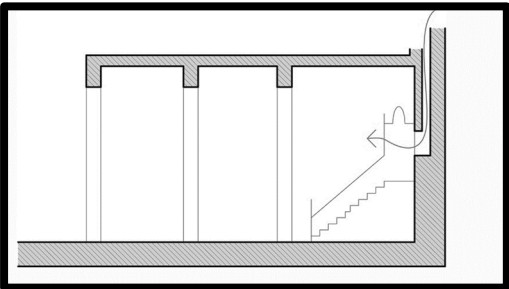

**Figure 12.** Section of the Mosque of al-Ṣāliḥ Ṭalā'i' at the minbar and the duct leading to the windcatcher, the arrow signifies the air flow through the duct and into the space (modified by the authors; based on [19,25]).

2.2.6. The Khanqah of Sultan Baybars Al-Jashankir (1306–1310)

The *Khanqah* (hostel for Sufis) of Sultan Baybars al-Jashankir had seven windcatchers [12]: one at the back of the side alcoves of the main *Iwan* (a vaulted portal opening onto a courtyard), one in the frontal niche of the opposite *Iwan*, and two in each of the *maglis'* back walls. The windcatcher in the western *Iwan*, the top of which is still accessible, offers additional information. A duct extends relatively high in the brick masonry and contains wooden beams, which are the remains of a canopy. A cornice halfway up the duct supports a horizontal element, allowing the windcatcher to be closed. The mausoleum, which was added later, features an equally interesting ventilation system. Two vertical ducts are embedded in the thickness of the dividing wall between the khanqah's west *Iwan* and the mausoleum (Figure 13). They lead to the roof and the alcoves on either side of the mihrab, each with its own wooden door leaf. There is cross ventilation between the vestibule of the mausoleum, which has large windows looking to the street, and a skylight on the roof, in addition to the ducts [19].

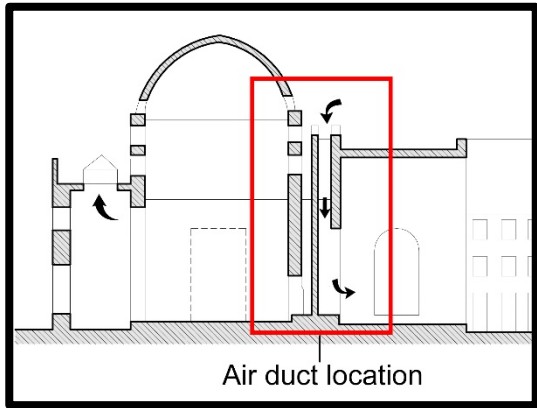

**Figure 13.** A longitudinal section of the Khanqah of Sultan Baybars al-Jashankir complex showing the ducts in the walls, the arrows signify the air flow through the space (modified by the authors based on [19]).

2.2.7. *Qã'a* of Muhibb Al-Din Ash-Shãf'i Al-Muwaqqi (1350)

To capture the ideal air volume and direct it downward, a windcatcher was installed in the northern Iwan [8] at *Qã'a* of Muhibb al-Din Ash-Shãf'i Al-Muwaqqi. The windcatcher on this building is the oldest example of a windcatcher made of stone that has survived for more than 500 years [12]. A model of the building (Figure 14), created by Hassan Fathy, shows the windcatcher on the left and the pavilion in the center [12,19]. The windcatcher's expansive canopy is open to the north and west but is closed to the south using a buttressed wall. The windcatcher was kept closed using wooden shutters. Moreover, through the windcatcher's opening in the ceiling, one can observe the amount of light that enters the space and illuminates the Iwan's front wall [19]. The air movement in the building is

illustrated in Figure 5, which shows the windcatcher supplementing the building with cool air while the skylight extracts the hot air.

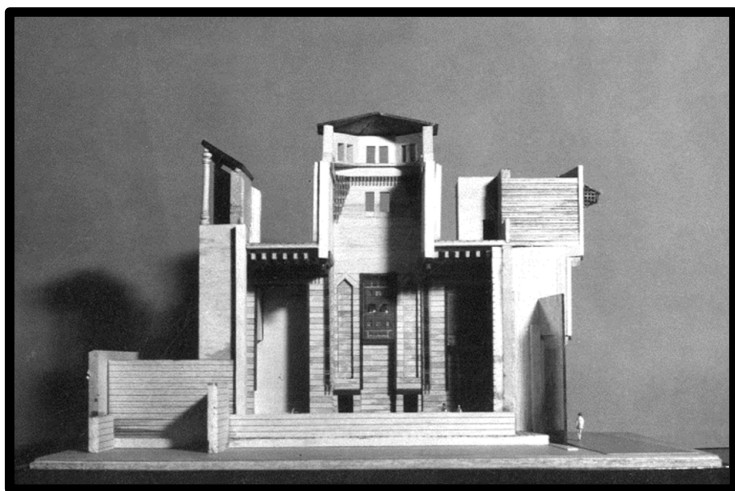

**Figure 14.** A 3D model of *Qa'a* of Muhibb al-Din al-Shāfi'i al-Muwaqqi. Reprinted with permission from [26]. 2023. Courtesy of the Rare Books and Special Collections Library, The American University in Cairo.

### 2.2.8. Suheimi House (1648)

The House of Suheimi includes a wooden windcatcher with a barred aperture toward the north (Figure 15). The roof of the windcatcher projects out in front of the aperture, and its form is typical of elaborate 19th-century architecture. In front of the windcatcher, a skylight topped with a small wooden dome on top of the *qa'a*'s entrance was designed to facilitate air circulation toward the *qa'a* [19]. The western side of the windcatcher should have been open as well or would have been before restorations [12]. This house had at least five *malqafs*, some on the second floor and in ruins, and a sixth one at the *qa'a* [12].

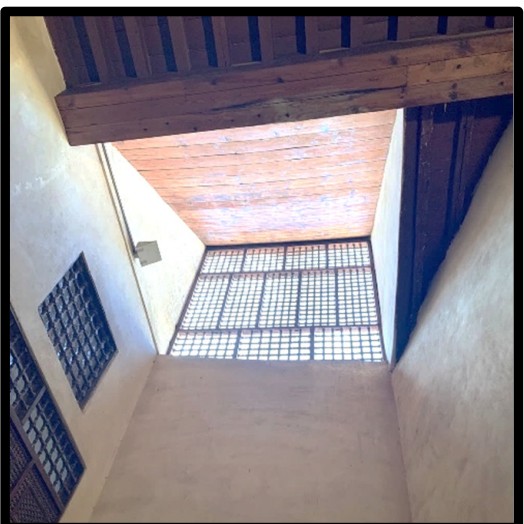

**Figure 15.** An interior view of the windcatcher of Bayt al-Suhaymi.

### 2.2.9. House of Alfi Bey (18th Century)

The courtyard at the House of Alfi Bey is dominated by three large windcatchers (Figure 16). The one on the left appears to be closed on the western side, whereas the eastern side of the one on the right is open. This contradicts the recommendations of medieval writings regarding a proper windcatcher opening [17].

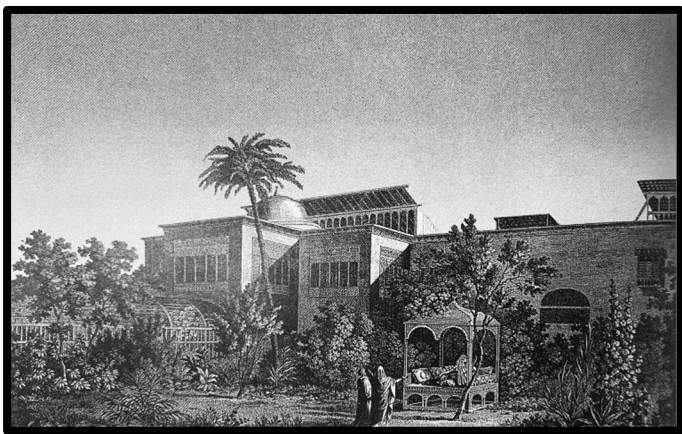

**Figure 16.** The exterior garden of the house of Alfi Bey is dominated by a decorative large windcatcher, 1821 [27]. This material is free to use, according to the Library of Congress in (see https://www.loc.gov/item/2021669727/ (accessed on 1 June 2023).)

### 2.2.10. Musāfirkhāne Palace (1779–1788)

The imposing windcatcher on the roof of Musāfirkhāne Palace survived until 1998. Its west side was open, as described in medieval astronomical sources [12]. The windcatcher was placed above the second floor on the antechamber preceding the *qa'a*. Moreover, the northern opening was protected using a rectangular roof projection [19], and the west side was open, thus agreeing with the medieval standards of windcatchers [12].

### 2.2.11. House of Al-Sinnārī (1794)

At the House of Al-Sinnārī, a large wooden canopy facing north constitutes the roof of the small *Iwan* south of the main *qa'a* (Figure 17). The northern and western openings are provided with frames with glazed windows. The *qa'a* has an unusual layout with the artificial separation established between this southern *Iwan* and the *durqa'a* using a high grille with a *Mashrabiya* (decorative projective windows prominent in Islamic architecture). This grille allows air to circulate from the windcatcher, through the *Iwan*, toward the dome on an openwork drum of the *durqa'a*. Dominated by the *malqaf*, this small *Iwan* appears as an independent room, which also acts as an antechamber [19].

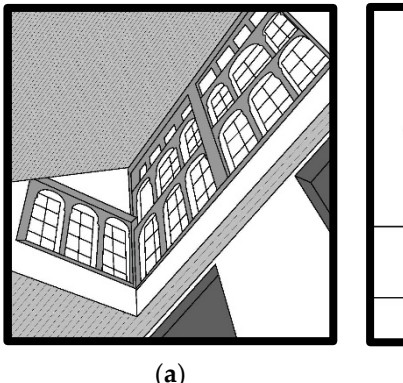
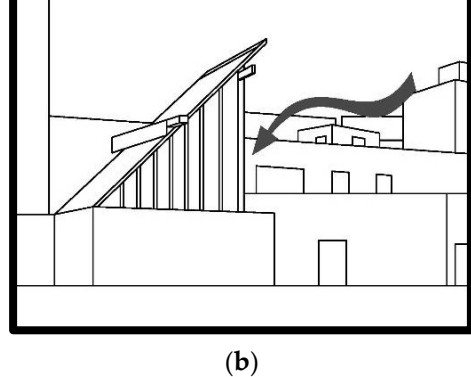

(**a**)  (**b**)

**Figure 17.** (**a**) An inside view of the windcatcher in the House of Al-Sinnārī. (**b**) An outside view of the windcatcher, the arrow signifies the air flow to the windcatcher (modified by the authors; based on [17]).

### 2.2.12. The Palace of Al-Jawhara in the Citadel (1814–1829)

All that seems to be left of Muhammad Ali's Palace's ventilation system is the single windcatcher shown in Figure 18 [12]. The awning of the windcatcher is on the ceiling of the central room, which connects to the peripheral rooms as well as the access staircase and the two sanitary and service blocks [19].

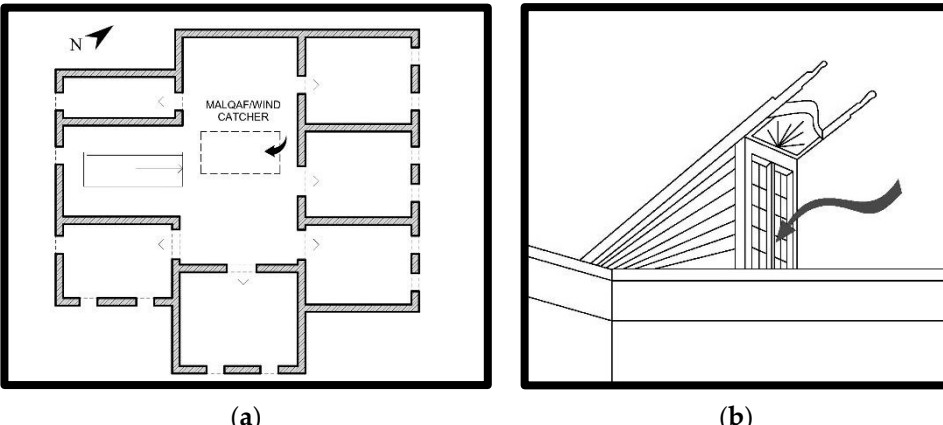

(**a**)                                                    (**b**)

**Figure 18.** (**a**) Plan for the Palace of al-Jawhara showing the location of the windcatcher (modified by the authors based on; [19]). (**b**) The northeastern elevation of the windcatcher, the arrows signify the air flow to the windcatcher (modified by the authors based on [19]).

*2.3. Modern Era in Egypt: Hassan Fathy's Designs, AUC Towers, and Sixth of October Villas*

Hassan Fathy and Ramsis Wissa Wassef were the most broadly known pioneer architects who revived vernacular architecture that used domes, vaults, and windcatchers. Hassan Fathy used the windcatchers in residential and public buildings such as the primary school for girls in El Gourna Village, where all the classes have an adjacent windcatcher facing north, from which the wind flows in and exits from another opening above the room. Placed inside the windcatcher of El Gourna is a sloped metal tray filled with wetted charcoal over which the air flows to cool the building. This object is similar to the *salsabil* that was used in the *Iwans* and halls of old Arab homes. Consequently, the windcatcher used in Gourna reduced the classroom's temperature by 10 degrees [15].

2.3.1. Hamdy Seif Al-Nasr House (1945)

Another building designed by Hassan Fathy that incorporated the use of windcatchers, more specifically, wooden ones, which is called Hamdy Seif Al-Nasr House, 1945 [28]. The windcatcher on the left of the dome directed airflow past a *salsabil* (cooling plate) that was kept damp using a continuous trickle from the earthenware water pot held above it (Figures 19 and 20) [16]. Since then, a roof stairway was added to the air shaft, changing its original design [28].

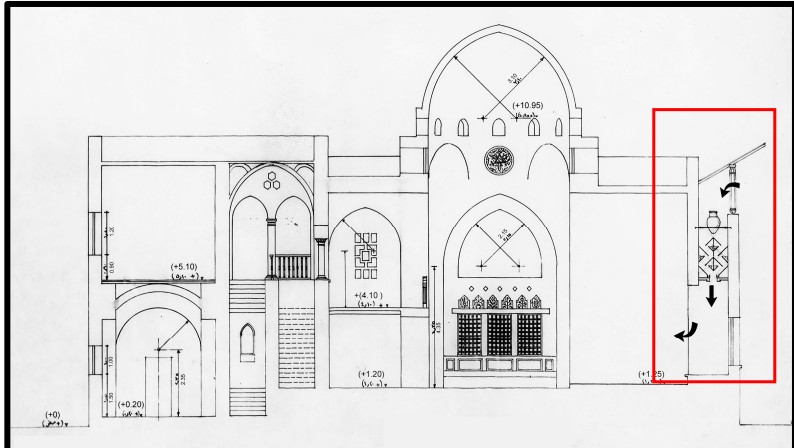

**Figure 19.** Section of Hamdy Seif Al-Nasr House, 1940s, the arrows signify the air flow from the wind catcher and into the space it ventilates and red boxes indicate the location of the wind catchers. Reprinted with permission from [29]. 2023. Courtesy of the Rare Books and Special Collections Library, The American University in Cairo.

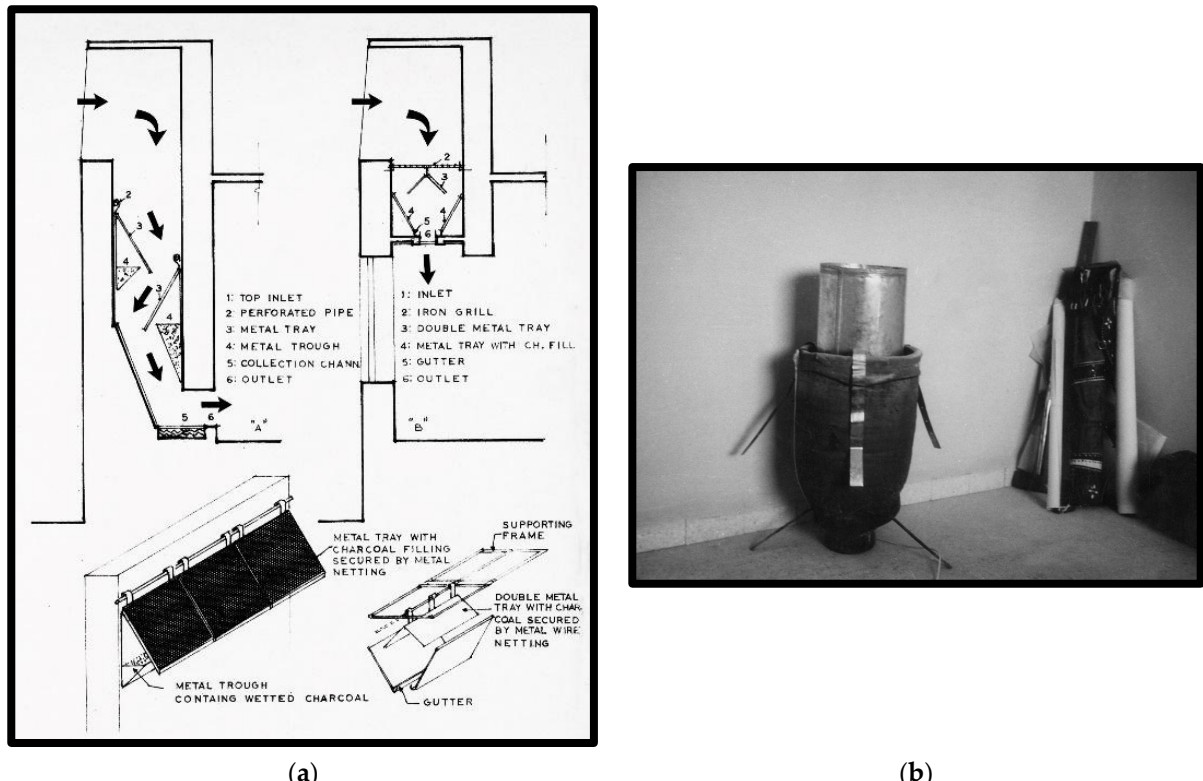

(**a**)　　　　　　　　　　　　　　　　　　　　　　　　(**b**)

**Figure 20.** (**a**) Sections 'A' and 'B' show the cooling mechanism used in the windcatcher of Hamdy Seif Al-Nasr House, the arrows signify the airflow. Reprinted with permission from [30]. 2023. Courtesy of the Rare Books and Special Collections Library, The American University in Cairo. (**b**) The earthenware water pot is used to drip water and provide evaporative cooling in the windcatcher used by Hassan Fathy. Reprinted with permission from [31]. 2023. Courtesy of the Rare Books and Special Collections Library, The American University in Cairo.

2.3.2. The Market of New Baris in El Wady El Gedid (1967)

Hassan Fathy also used a windcatcher in the market of the New Baris Oasis, where the windcatcher directs the wind to flow in the basement where all biodegradable fresh produce is stored for the sake of preservation [16]. The planning, sectioning, and elevation of the market of New Baris in El Wady El Gedid ensure that wind is drawn inside using a row of windcatchers. The wind is then directed into the basement, where the goods susceptible to spoiling are stored (Figures 21 and 22).

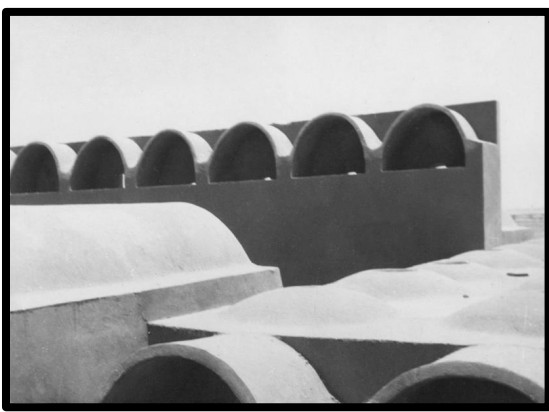

**Figure 21.** The market of New Baris in El Wady El Gedid, 1960s. Reprinted with permission from [32]. 2023. Courtesy of the Rare Books and Special Collections Library, The American University in Cairo.

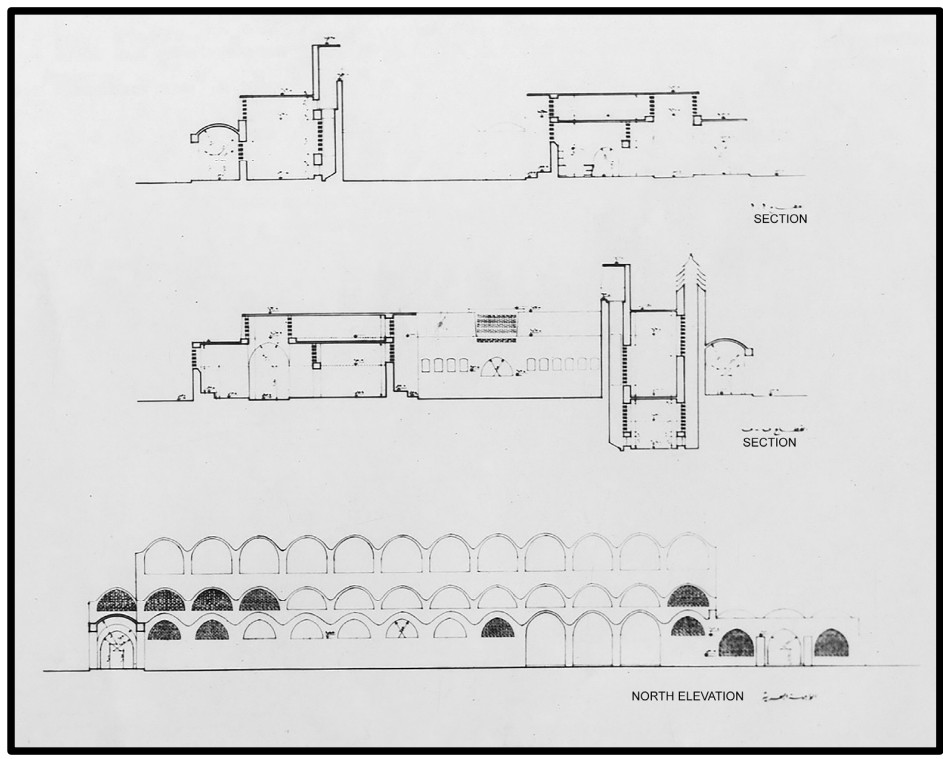

**Figure 22.** Sections and elevation of the market of New Baris in El Wady El Gedid, 1960s. Reprinted with permission from [33]. 2023. Courtesy of the Rare Books and Special Collections Library, The American University in Cairo.

2.3.3. The Luxor Cultural Center (1970)

The Luxor Cultural Center was a complex designed by Hassan Fathy in 1970 for the Egyptian Ministry of Culture. The center was planned to be in the Middle of Luxor, close to the Sidi al-Wahsh Fatimid Mosque. However, only the main hall was finished. Moreover, the architect's plans for natural ventilation were disregarded. The entire interior space was to be cooled using a large windcatcher [34]. As seen in the sections in Figure 23, Fathy's design of the windcatcher was open on opposite sides: one opening was the inlet, which faced northeast, and one was the outlet that let the hot air escape, facing southwest. This iteration of the windcatcher seems to have been an experimental approach for the *malqaf* by Hassan Fathy, as seen in the Experimental Rooms for the Ministry of Scientific Research designed by the architect (Figure 24). Unfortunately, the envisioned windcatcher was closed off, making the roof form meaningless [35].

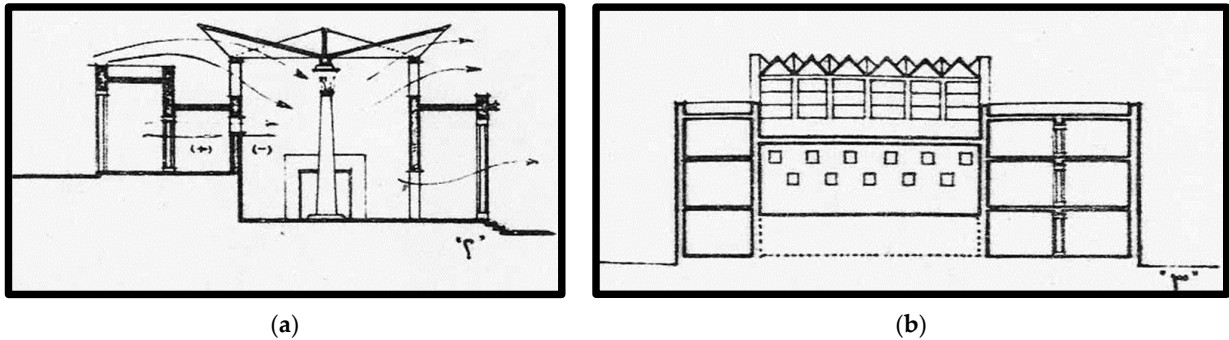

(**a**)                                                                 (**b**)

**Figure 23.** (**a**,**b**) Sections of Luxor Cultural Center by Hassan Fathy, 1960s, the arrows signify the air flow. Reprinted with permission from [35]. 2023. Courtesy of the Rare Books and Special Collections Library, The American University in Cairo.

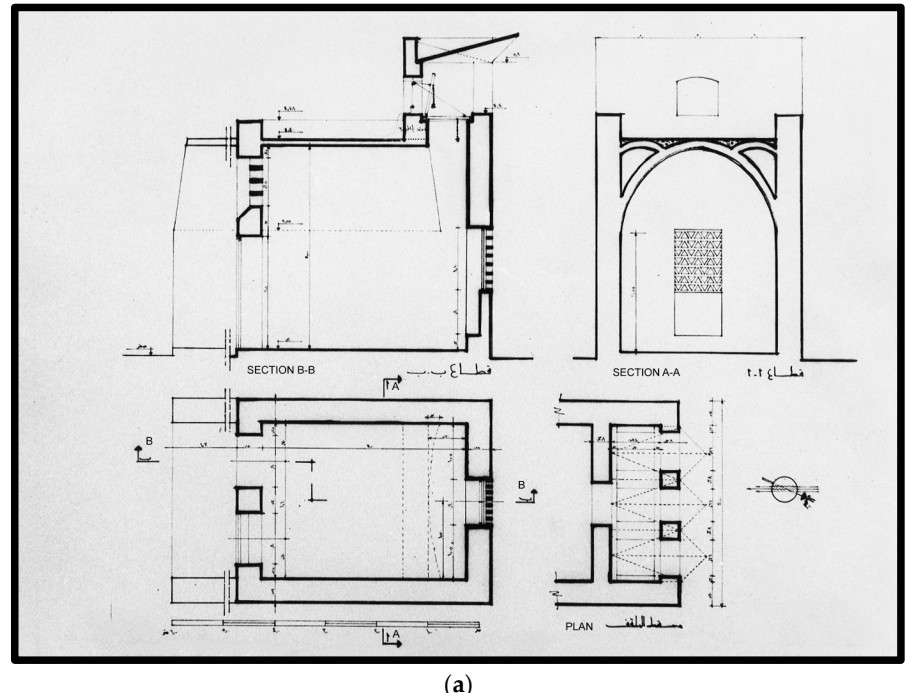

(**a**)

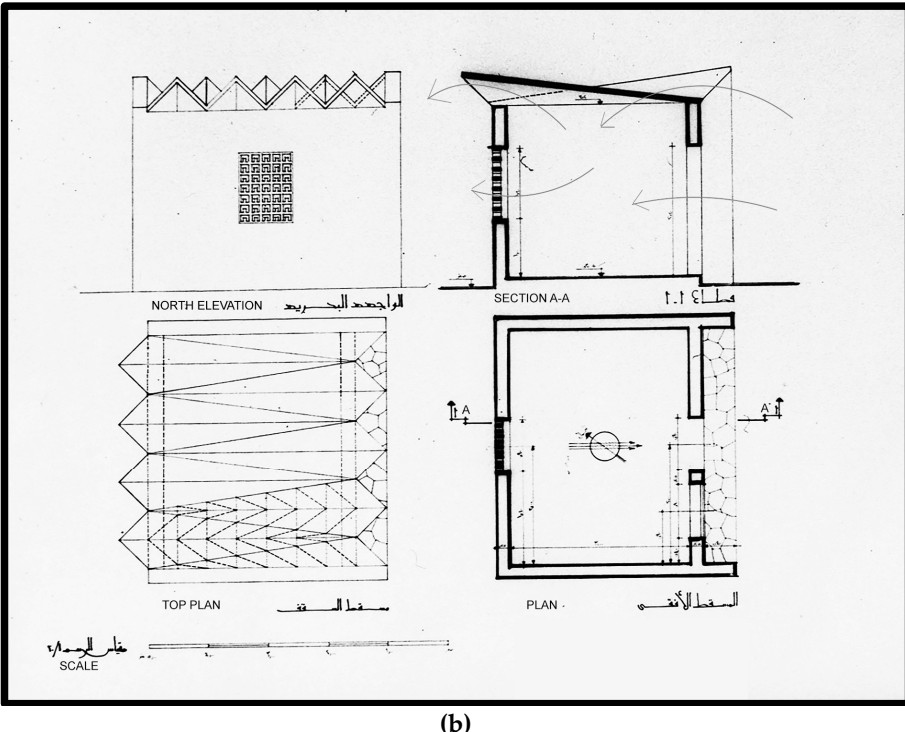

(**b**)

**Figure 24.** (**a**,**b**) Sections and plans for the Experimental Rooms for the Ministry of Scientific Research by Hassan Fathy, the arrows signify the air flow from the wind catcher and into the space it ventilates. Reprinted with permission from [36]. 2023. Courtesy of the Rare Books and Special Collections Library, The American University in Cairo.

### 2.3.4. A Modern Apartment Building

The design for a modern apartment building by Hassan Fathy shows the use of a staircase as a *malqaf* with a cooling mechanism based on evaporation on top of the stairwell. From the plan and section in Figure 25, the windcatcher is shown to be facing south, which is different from the typical orientation of the windcatcher in Egypt.

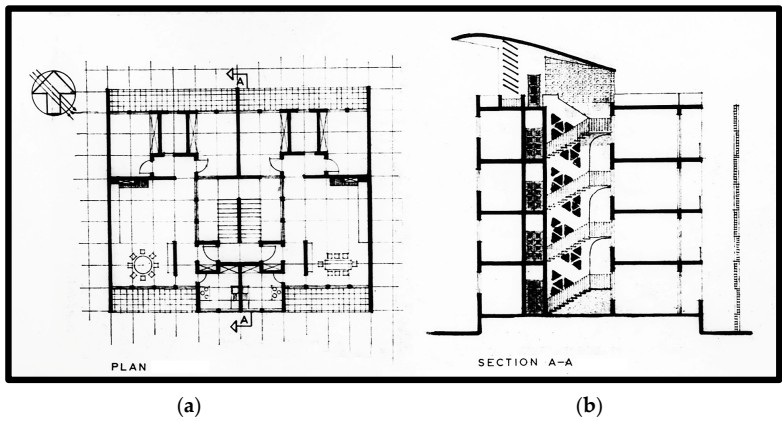

|  |  |
|---|---|
| (**a**) | (**b**) |

**Figure 25.** (**a**,**b**) Plan and section 'A-A' for a modern apartment building by Hassan Fathy. Reprinted with permission from [37]. 2023. Courtesy of the Rare Books and Special Collections Library, The American University in Cairo.

2.3.5. Villa "A" in 6th of October City

Architect Ramy El Dahan was one of Hassan Fathy's students. Continuing his mentor's legacy, he designed many public and residential buildings using elements of vernacular architecture. Among the buildings he designed is "Villa A" (See Figures 26 and 27) in El Thawra El Khadra, located next to El Sheikh Zayed on the 6th of October City. It was designed using a courtyard, vaults, and domes, and he used the windcatcher to provide cool natural ventilation for the whole villa. The windcatcher, located on the villa's ground floor, connects to the basement. This allows the air to circulate and drop more of its temperature. As seen in Figure 27, some of the air shafts are also embedded in the walls. The main function is to pull the air that has been circulating underground to cool down the basement (see Figure 26) to the ground and upper two floors. These shafts achieve this cooling using small openings in the walls of the rooms. Overall, this process helps cool down the air in most indoor spaces, and the villa's ability to operate successfully without mechanical air conditioning units is evidence of this method's success [38].

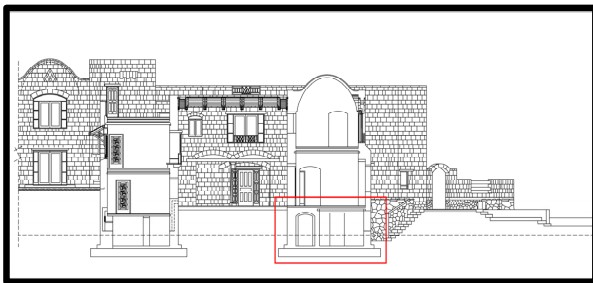

**Figure 26.** Villa A section showing the basement, in the red square, where the wind from the wind catcher is directed and cooled down.

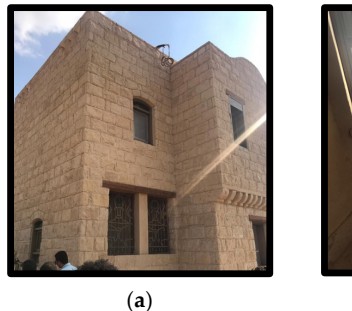 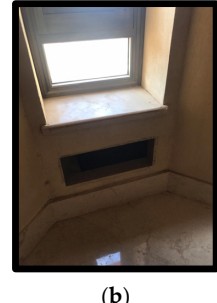

|  |  |
|---|---|
| (**a**) | (**b**) |

**Figure 27.** (**a**) Windcatcher, and (**b**) air shaft.

### 2.3.6. Villa "B" in 6th of October City

Another successful modern example of a residential building that uses a windcatcher is a villa designed by Prof. Dr. Ahmed Reda Abdeen located in 6th of October City (Figure 28a). This villa also utilizes passive design techniques for cooling. The villa consists of three floors in addition to the basement. The windcatcher that is studied in this villa rises above the roof by 1.5 m and is oriented north-westward (Figure 28b). It works alongside a solar chimney, which is painted black to further enhance the natural ventilation in the villa's indoor spaces (Figure 29). Both features are connected using underground tubes, as can be seen in Figure 29c. Because the solar chimney is painted black, the air flowing inside it is at a higher temperature due to the color's high absorption of heat. This forced increase in temperature causes the unwanted hot air to be vented upward faster. This hot air is then replaced with cooler air coming from the windcatcher. As in Villa A, this windcatcher also pulls the outdoor air into the underground tubes where it circulates to cool down, and then it is led upward to be provided inside the rooms. This process successfully cools the rooms in the villa without the need for mechanical cooling [38].

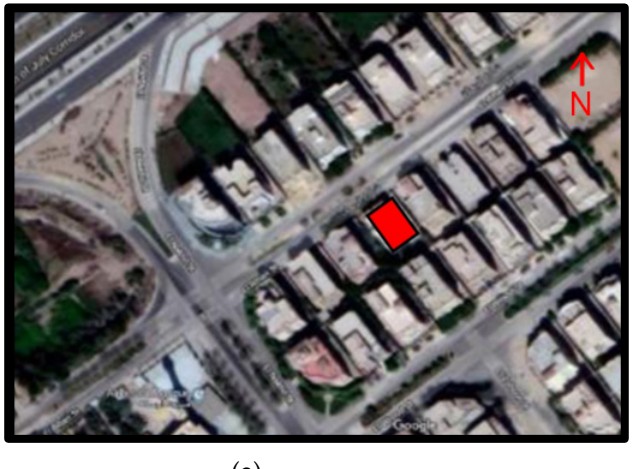

(**a**)

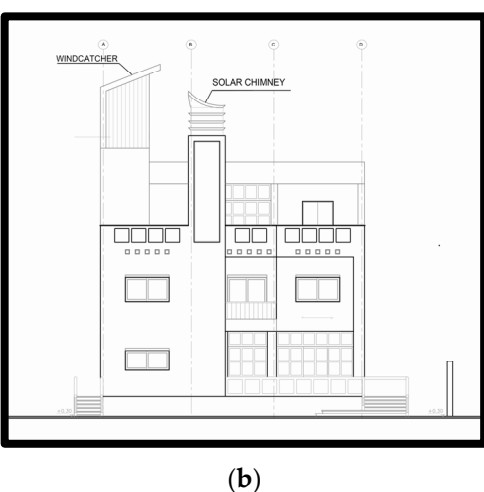

(**b**)

**Figure 28.** (**a**) Villa B layout (in red) (modified by authors, adapted from [38]). (**b**) Villa B elevation.

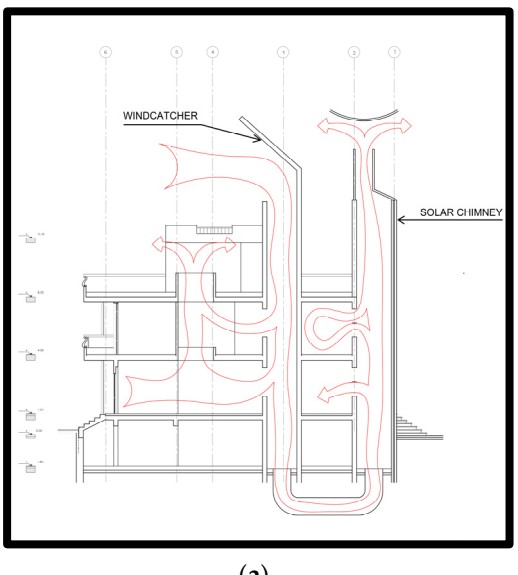

(**a**)

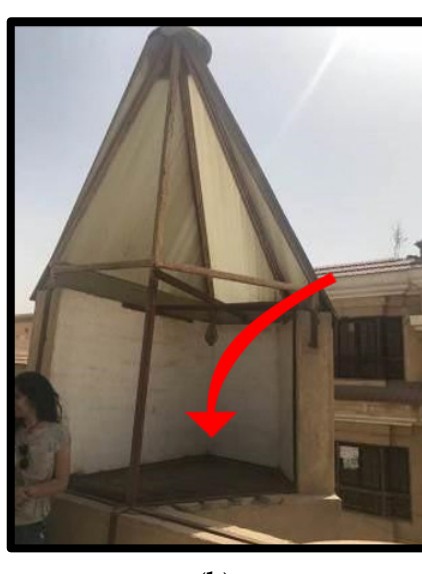

(**b**)

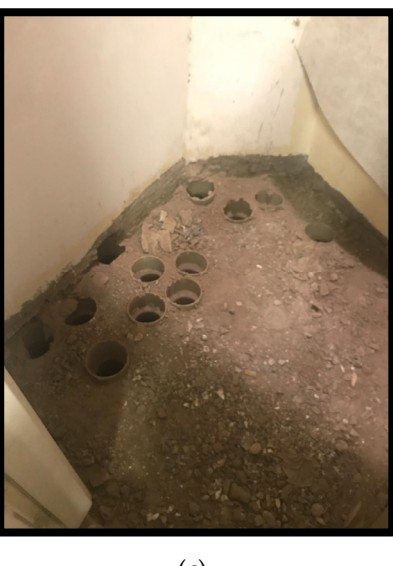

(**c**)

**Figure 29.** (**a**) Section showing the windcatcher and solar chimney. (**b**) View of the wind catcher (modified by authors, adapted from [38]). (**c**) View of the underground tubes in the windcatcher.

### 2.3.7. The American University in Cairo New Campus (2008):

Finally, the AUC campus master planned by CDC Abdel Halim and Sasaki and Associates, specifically, the School of Sciences and Engineering (SSE), which was designed by Sasaki and Associates, offers two full-scale prototypes of windcatchers [39]. Both towers face the same direction, as shown in Figure 30, and are located over semi-enclosed atria that open on courtyards and have a similar configuration, where both are four-sided square plans with interior cross partitions. However, they differ in size (height and side length) and interior and exterior finishes. Moreover, Tower A is a straight-tower chimney with fixed metal louvers (Figure 31a), while Tower B features an inclined tower with operational louvers (Figure 31b). Both towers provide natural ventilation to the connected spaces, which decreases the use of mechanical cooling.

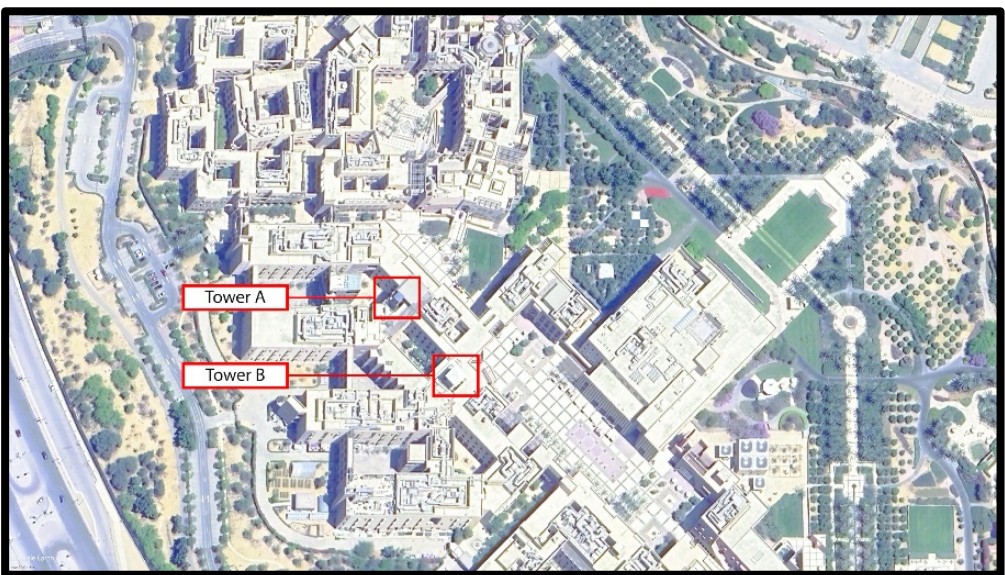

**Figure 30.** Site plan for the SSE building at AUC (obtained from Google Earth and adapted by the authors).

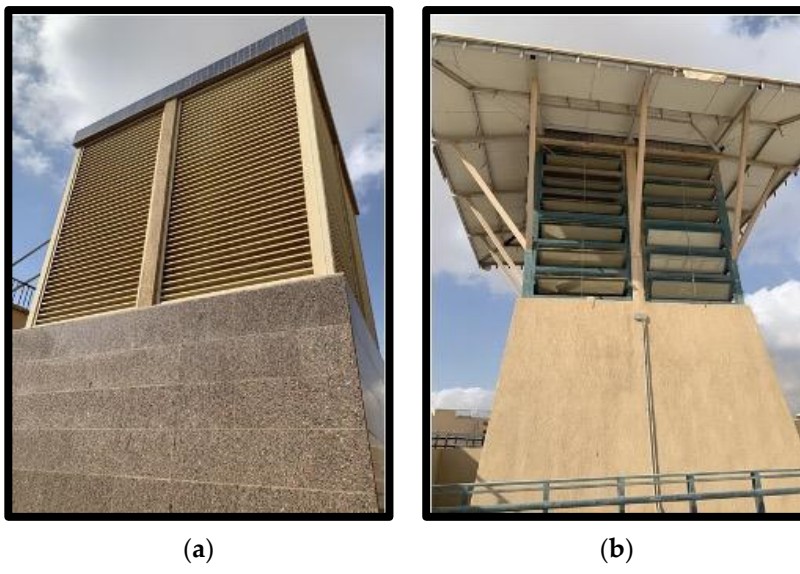

|  |  |
|:-:|:-:|
| (**a**) | (**b**) |

**Figure 31.** (**a**) Tower A is a straight-tower chimney with fixed metal louvers and (**b**) Tower B is an inclined tower with functional louvers.

### 3. Case studies from the Middle East (Old and New)

*3.1. Historical Case Studies from the Middle East*

3.1.1. The Tower of Dolatabad Windcatcher, Iran

Acknowledged as a world heritage site by UNESCO, the Dolatabad Garden in Yazd is home to Iran's tallest wind turbine with a 34 m height, which was rebuilt after it collapsed in the 1960s. The eight-sided wind tower is used to ventilate the building by sucking the air that flows inside the building and passing it over a small rocky pool through the water jet. It is then is channeled to other rooms, as seen in Figure 32 [40].

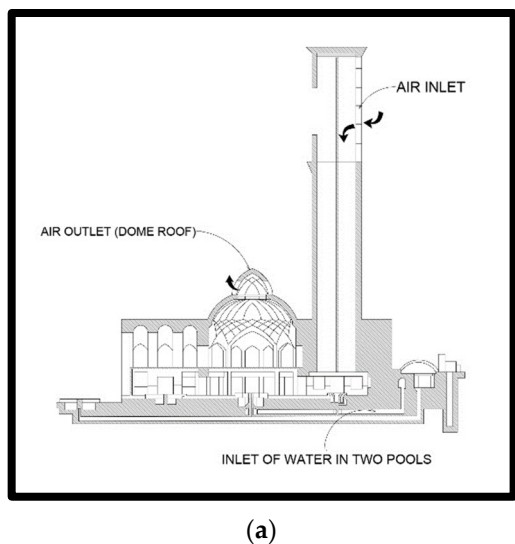 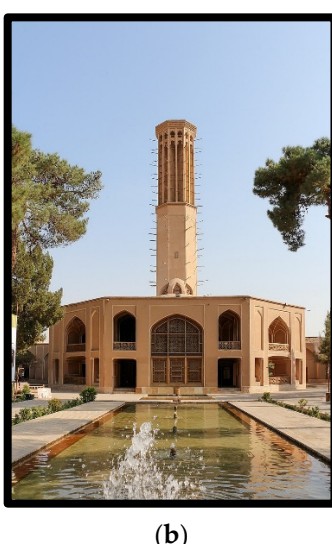

(**a**)  (**b**)

**Figure 32.** (**a**) A section through the building and windcatcher (modified by the authors; adapted from [40]). (**b**) Photograph showing the tower by Bernard Gagnon under a GNU Free Documentation License.

3.1.2. The Ganjali-Khan Square Windcatcher, Iran (1596–1621)

The Ganjali-Khan complex in Iran consists of a wide rectangular square with 50 × 100 m dimensions (Figure 33). This square comprises important public and semi-public buildings such as a caravanserai on the eastern side, a mosque on the northeastern side, a bathhouse to the south, a water reservoir and school to the west, and a mint to the north. The windcatcher, the tallest element in the square, is located near the north entrance and is a four-sided square shape decorated with brick and tilework. It is attached to the single-floor arch lobby connected to the courtyard (Figure 33b) [41].

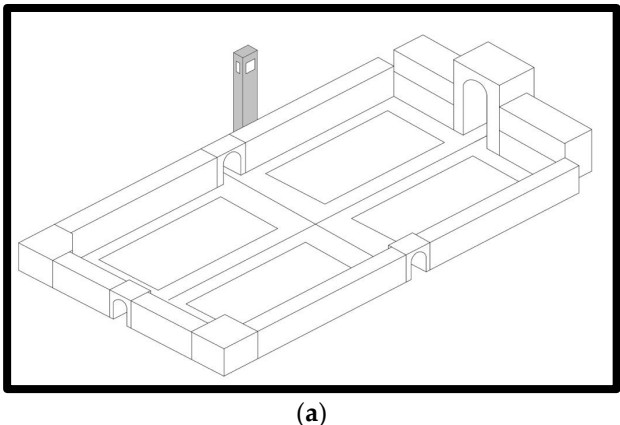 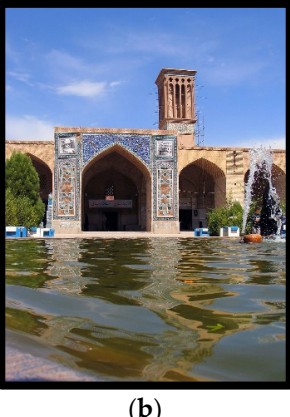

(**a**)  (**b**)

**Figure 33.** (**a**) A 3D isometric drawing showing the placement of the windcatcher in the Ganjali-Khan Square (recreated by the authors; based on [41]). (**b**) Photograph showing the tower by Cyrus the Great under a Creative Commons Attribution 2.5 Generic license.

### 3.1.3. Bastakiya Windcatchers, Dubai, UAE (1890)

The Bastakiya windcatchers were built by Persian immigrants who migrated from Iran to Bastak in Dubai [20,42]. The immigrants started to construct structures like those in Iran, where the climate is warm and humid. Furthermore, windcatchers in Bastakiya, Dubai, are four-sided and usually square in a plan of approximately 2.5 × 2.5 m² [20]. Figure 34 shows photographs of the four-sided Dubai windcatchers. Moreover, Figure 35 shows the possible air movement in a windcatcher in Dubai as analyzed by Hassan Fathy [43].

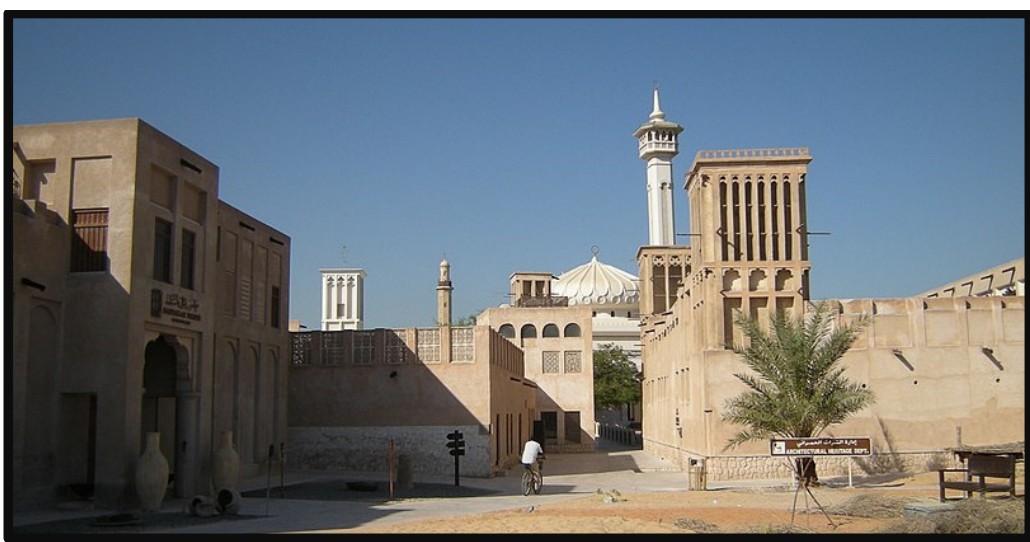

**Figure 34.** Four-Sided windcatchers in Bastaqyia, Dubai, UAE. modified by authors, original photograph by Russavia under Creative Commons Attribution 2.0 Generic.

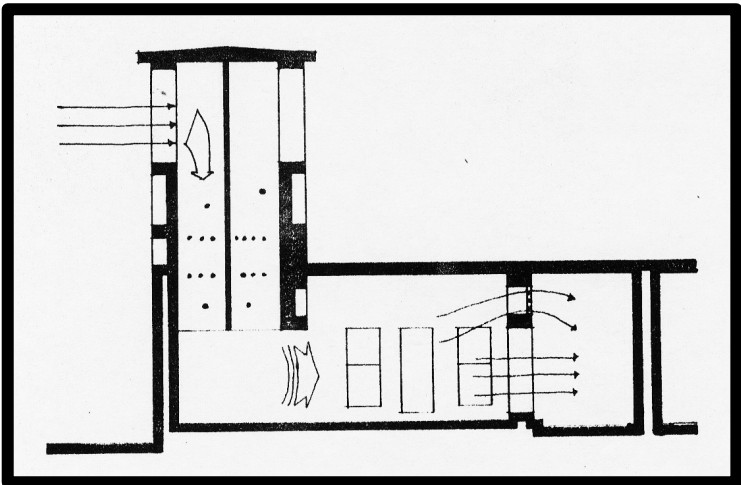

**Figure 35.** Air movement in a wind tower in Dubai that was designed by Hassan Fathy in the 1970s. Reprinted with permission from [43]. 2023. Courtesy of the Rare Books and Special Collections Library, The American University in Cairo.

### 3.1.4. Diwan-Khanat Al-Asterabadi, Baghdad, Iraq (19th Century)

Iraqi architect Dr. Subhi Al-Azzawi documented the windcatchers in the Diwan-Khanat al-Asterabadi, Baghdad, Iraq, where there are four windcatchers. Three face the northeastern wind, and a larger one faces northwest (Figure 36a). Furthermore, the scoops of the windcatchers face inwards to the building rather than outwards. The unidirectional windcatcher (Figure 36b) reaches through the floors of the building and as far as the basement. Moreover, to aid in cooling the wind, a sizeable baked clay jug (*zir*) of water is placed at the base of the column [12].

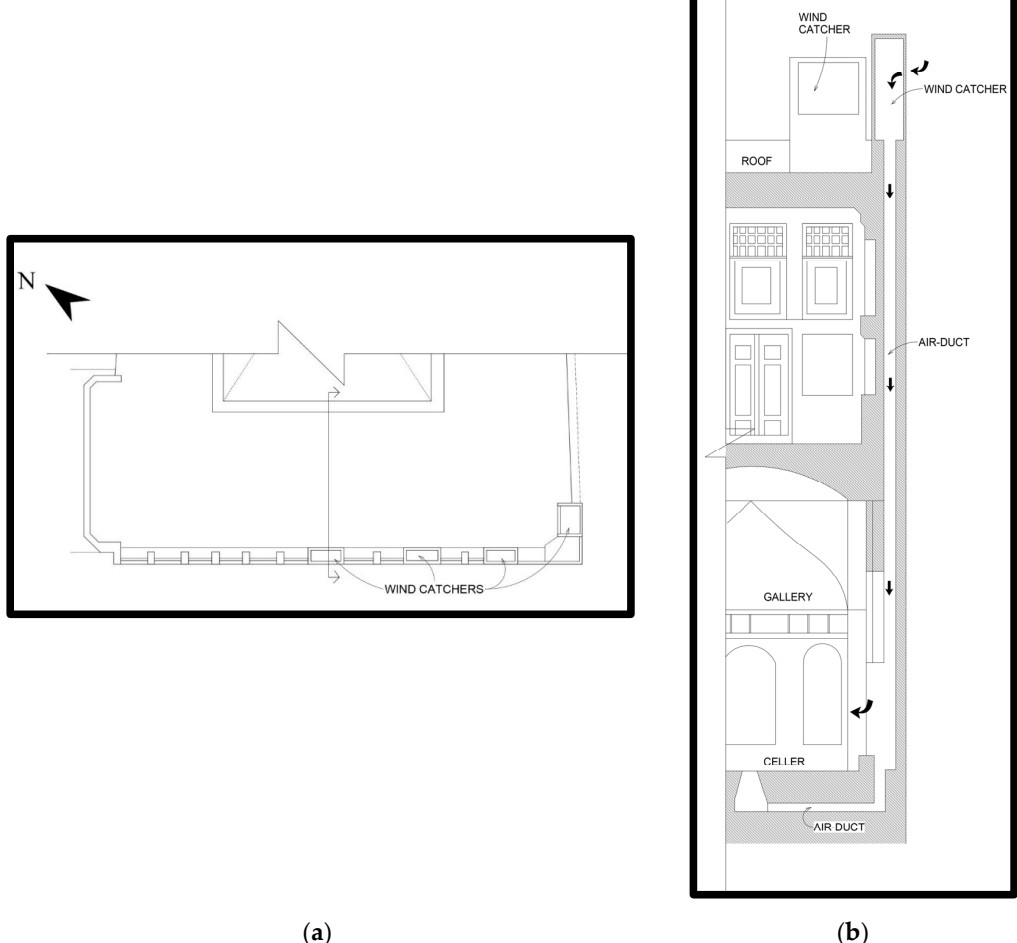

(**a**) (**b**)

**Figure 36.** (**a**) Plan and (**b**) section showing four windcatchers in Diwan-Khanat al-Asterabadi, Iraq (modified by the authors; adapted from [17]).

Moreover, windcatchers in Iraq are usually rectangular in cross-section. The column of a typical windcatcher in Iraq varies from 15 to 60 cm in width, and its height never exceeds the roof of its building but may begin from the roof. The roof of a windcatcher is inclined at 45°. A windcatcher is usually 2 m from the roof. Columns usually end in the basement of the building, where the air flows through small metal windows underground [44].

*3.2. Modern Cases from the Middle East*

3.2.1. Masdar City, Abu Dhabi, UAE (2010)

Despite the fact that Masdar City is located in an area with moderate wind, the city's planners decided to place the wind tower in the Public Square at the Institute of Science and Technology in Masdar City, rather than in a home or other enclosed space, to have a greater impact on more populated areas. At the base of the wind tower is a sizable urban plaza that is home to a variety of purposes, including cafes and other retail establishments, as well as a seating area that enjoys beautiful weather. The thermal performance of the enormous windcatcher in the public metropolitan area, which depends on modern technology, is shown in Figure 37. A stage is elevated beneath the windcatcher and is used for performance. Sensors manage the automatic louvers on the windcatcher. These sensors keep track of the predominant wind direction and control the louvers to send the wind down the tower in the appropriate direction. As an evaporative cooling system, the tower also has a high-temperature mist jet to humidify the incoming air and make it cooler at ground level [45].

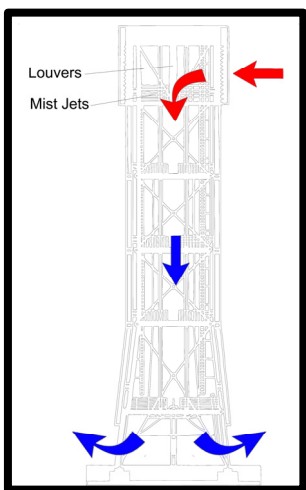

**Figure 37.** A section through the windcatcher that shows its performance, the red arrow signifies hot air, while the blue shows that the air has been cooled (modified by the authors; adapted from [45]).

3.2.2. Khalifa Stadium, Doha, Qatar (2017)

Another example of a building using windcatchers as a cooling tool is the Khalifa Stadium, which was built for the FIFA 2022 World Cup in Qatar (Figure 38). The cooling system used by the fans is installed in cooling towers located more than 1 km away from the stadium. The cooled water is pumped from the towers into the stadium. There are 9 units in the NCR2610 fan, with a 26 ft diameter, and 10 blades with an airflow capacity of 433.4 m$^3$/s [46].

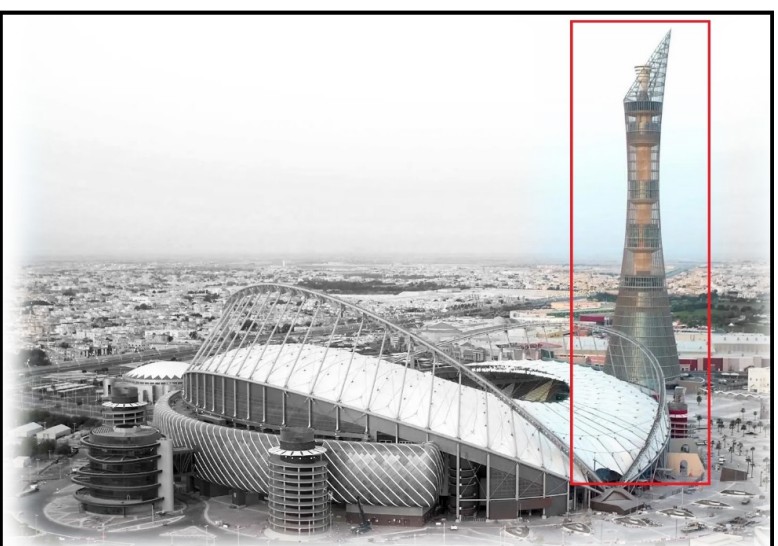

**Figure 38.** Khalifa International Stadium showing the wind tower in the red square; adapted from [47].

The Khalifa International Stadium is the largest cooled outdoor stadium in the world. A system that works to reduce energy consumption was therefore used in this building. This system, called "urban refrigeration, "successfully keeps the players and the audience within the comfort range. In this system, refrigeration technology is 40% more sustainable than conventional techniques. This is made possible with the utilization of an energy center located one kilometer from the stadium. In this center, the cooled water is carried through a pipe to the venue. The cold water then arrives, cools the air, and then is pushed toward the center of the stadium, the playing field, and the seating areas to reach a total of about 500 vents dispersed around the stands. Moreover, because the cold air is denser, it remains

low near the ground; therefore, not all the hot air enters [46]. This is accomplished by the effect of buoyancy force, as discussed in the Introduction section of the paper.

## 4. Results of the Case Study Analysis

The tables below contain a summary of the cases investigated in this study. Windcatchers started showing up during the Pharaonic and Coptic eras, but the shape, the number of sides, orientation, and openings seem to have been developed further during the Islamic era, when there was more understanding of wind movement and direction prevalence (see Figure 6) [12,17]. Table 2 shows that 85% of the windcatchers in Egypt are either single-sided or two-sided, and most are oriented toward northern and western directions. This study consisted of 12 case studies from the Pharaonic to the Coptic to the Islamic eras. Overall, 50% of the cases of windcatchers studied from the aforementioned eras are single-sided. Moreover, 33.3% of them are two-sided, all of which have their openings adjacent to each other, and all of which are from the Islamic era. In total, 75% of the two-sided windcatchers face north and west, which is the span of the prevalent wind directions in Egypt (see wind rose in Figure 39a). Furthermore, there are cases for which the type and orientation of the wind towers are unknown since they no longer exist [10].

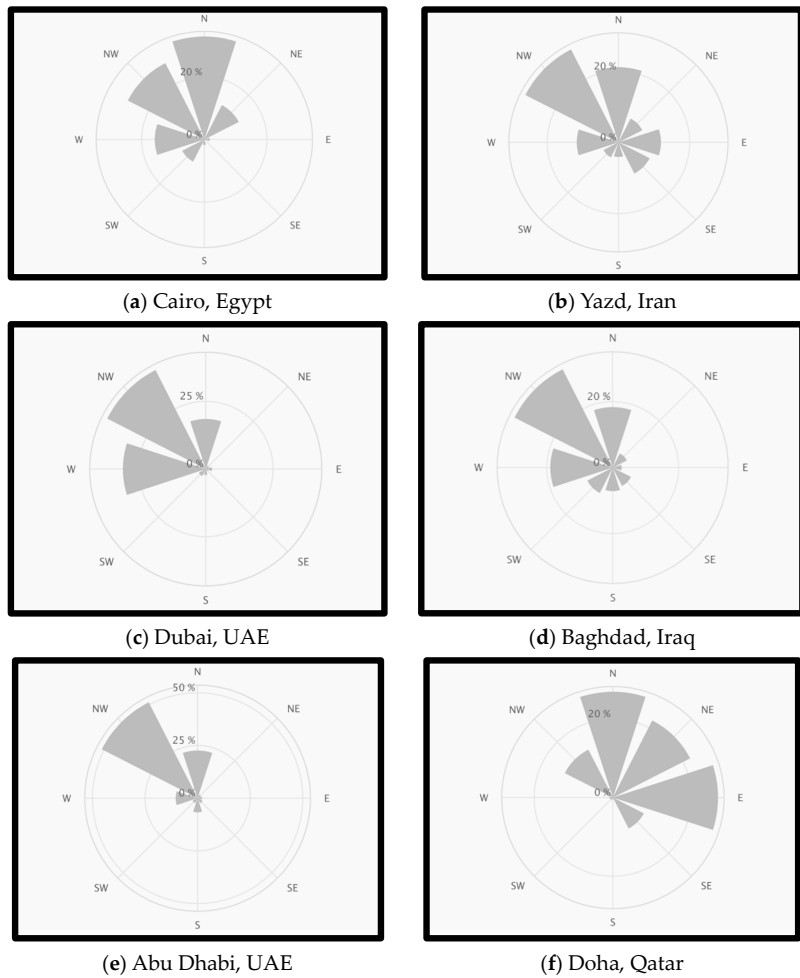

(**a**) Cairo, Egypt

(**b**) Yazd, Iran

(**c**) Dubai, UAE

(**d**) Baghdad, Iraq

(**e**) Abu Dhabi, UAE

(**f**) Doha, Qatar

**Figure 39.** Wind roses in the regions studied. (Note: wind roses are available at the website: World-Weather Weather Archive in Yazd Available online: https://world-weather.info/archive/ accessed on 16 April 2023/).

The table further explains how windcatchers changed from the Medieval to the Modern era. Overall, 37.5% of the modern case studies have windcatchers that follow the design standards of vernacular windcatchers. Nonetheless, it seems that Hassan Fathy developed the *malqaf*, as seen in the cases of the Luxor Cultural Center and Experimental

Rooms for the Ministry of Scientific Research, where the openings of the windcatchers face opposite directions. A study examining various windcatcher design parameters as a means for the passive cooling of low-rise buildings in Egypt during the summer shows that an iteration similar to Hassan Fathy's resulted in the highest air change per hour [48]. For example, none of the ancient or medieval windcatchers are flat-roof, and 37.5% of the modern windcatcher case studies have flat roofs. The effectiveness of modern flat roof windcatchers is yet to be studied in Egypt's climatic conditions.

Table 3, on the other hand, indicates that wind towers throughout the Middle East varied in type, ranging from single-sided to cylindrical types. It is worth noting that both modern wind towers studied are cylindrical. The shapes found in the tables below align with the findings in Table 1 [9].

The following wind roses in the various studied regions determine if the windcatchers studied here considered the prevalence of the wind. Cairo's wind rose shows that the prevailing wind comes primarily from the north and northwest. More than 32% of the wind appears from the north, followed by more than 27% from the northwest. No wind whatsoever comes from the span between the south and east, and negligible amounts of wind are received from the remaining directions (Figure 39a). This explains why wind towers in Cairo are typically single-sided or two-sided. It would be useless, and maybe even harmful to the speed of the wind coming in, to have many unnecessary openings in the tower's design when only one general direction is receiving most of the wind.

Like Cairo, Yazd's wind rose shows that most of the wind comes from the northwest and north. More than 29% of Yazd's wind appears to come from the northwest. This is followed by slightly more than 21% from the north. On all other sides, very little wind is received (between 4 and 11 percent) (Figure 39b). This explains why wind tower designs in Iran vary in their number of sides, as all sides receive wind, even if very little, as seen in the examples above for the tower of Dolatabad and the Ganjali-Khan Square windcatchers. Therefore, the designs depend on each area's specific prevailing wind directions.

In Dubai, the wind rose indicates that the wind comes primarily from the northwest and west. Slightly more than 40% of the wind in Dubai comes from the northwest. This is followed by more than 30% coming from the west. Almost no wind at all comes from the span between the southwest and northeast (only between 0.9 and 2.1 percent) (Figure 39c). Even though there seems to be a prevalent wind direction (northwest) in Dubai, the windcatchers found in Bastaqyia, Dubai, UAE are four-sided, which might seem contradictory to the typical use of one-sided windcatchers for places that have a prevalent wind direction.

The prevailing wind in Iraq is generally from the northwest, comprising 33.4%, followed by the west and the north follow, with each receiving approximately 18% of the wind. Based on the information on the wind rose shown in Figure 39d, single-sided windcatchers, just like in Egypt, are understandably used in Iraq, as the above example of Diwan-Khanat al-Asterabadi shows. However, the direction of the particular windcatcher that faces northeast does not align with the wind rose.

It is evident from Abu Dhabi's wind rose that the prevailing wind direction is northwest. It receives notable 51.1% of all the wind in Abu Dhabi. This is followed by less than half that amount received in the north (only 22.6%). All the other sides receive very trivial amounts of wind (between 1.8 and 10 percent) (Figure 39e). This can explain why the wind tower designs in this region vary, along with the previous wind rose from the Emirates. There is a high level of variance in the wind statistics from one region to the next; therefore, the design also varies in response to the wind statistics.

Lastly, Doha's wind rose indicates several diverse prevailing wind directions. This is because the differences between them are minimal. Doha receives 26.9% of its wind from the north, 26.7% from the east, and 22% from the northeast. The northwest direction receives slightly less, at almost 14%, and the southeast direction is slightly lower, at almost 9%. The rest of the sides receive negligible amounts of wind (<1%) (Figure 39f). This explains why there is diversity in the wind towers, which typically have many sides.

**Table 2.** Types of windcatchers throughout different eras in Egypt.

| | Building | Type of Building | Historic Era | Type of Tower | | | | | Orientation of Openings | Shape |
|---|---|---|---|---|---|---|---|---|---|---|
| | | | | Single Sided | Two-Sided | | Four-Sided | Unknown | | |
| | | | | | Opposite | Adjacent | | | | |
| 1 | Neb-Amun | Residential | Pharaonic | √ | | | | | Unk | Sloped roof |
| 2 | Virgin Mary Hanging church | Religious | Coptic | √ | | | | | NE | Sloped roof |
| 3 | Virgin Mary Al-Damshareya church | Religious | Coptic | √ | | | | | Unk | Sloped roof |
| 4 | Mosque of al-Ṣāliḥ Ṭalā'iᶜ | Religious | Islamic | √ | | | | | N | Unk |
| 5 | Madrasa of al-Nāṣir Muḥammad | Educational | Islamic | | | | | √ | Unk | Unk |
| 6 | Khanqah and Mausoleum of Sultan Baybars al-Jashankir | Complex | Islamic | | | | | √ | Unk | Unk |
| 7 | Qāᵓa of Muhibb al-Din Ash-Shāf'I Al-Muwaqqi | Residential | Islamic | | | √ | | | N and W | Sloped roof |
| 8 | Suheimi House | Residential | Islamic | √ | | | | | N | Sloped roof |
| 9 | Three windcatchers on the House of Alfi Bey | Residential | Islamic | | | √ | | | N and E | Sloped roof |
| 10 | Musāfirkhāne Palace | Residential | Islamic | | | √ | | | N and W | Sloped roof |
| 11 | House of Al-Sinnari | Residential | Islamic | | | √ | | | N and W | Sloped roof |
| 12 | The Palace of al-Jawhara in the Citadel | Residential | Islamic | √ | | | | | N | Sloped roof |
| 13 | Seif Al-Nasr House | Residential | Modern | | | √ | | | Ukn | Sloped roof |
| 14 | New Baris Oasis Market | Commercial | Modern | √ | | | | | N | Flat roof |
| 15 | Luxor Cultural Center | Complex | Modern | | √ | | | | NE and SW | Sloped roof |
| 16 | Experimental Rooms by Hassan Fathy | - | Modern | | √ | | | | NW and SE | Sloped roof |
| 17 | Modern Apartment Building | Residential | Modern | √ | | | | | S | Sloped roof |
| 18 | Villa A in 6th of October City | Residential | Modern | √ | | | | | NW | Flat roof |
| 19 | Villa B in 6th of October City | Residential | Modern | √ | | | | | NW | Sloped roof |
| 20 | School of Sciences and Engineering (AUC) | Educational | Modern | | | | √ | | NW, NE, SW, ad SE | Flat roof |

√ = Yes

**Table 3.** Types of windcatchers throughout different eras in Middle East.

| | Building | Type of Building | Historic Era | Country | Type of Tower | | | | Orientation of Openings | Shape |
|---|---|---|---|---|---|---|---|---|---|---|
| | | | | | Single Sided | Four-Sided | Eight Sided | Cylindrical | | |
| 1 | Tower of Dolat Abad | | Islamic | Iran | | | √ | | All | Flat roof |
| 2 | Ganjali-Khan Square | | Islamic | Iran | | √ | | | N, E, S, andW | Flat roof |
| 3 | Bastaqyia | Residential | 19th century | United Arab Emirates | | √ | | | - | Flat roof |
| 4 | Four windcatchers on Diwan-Khanat al-Asterabadi | Residential | 19th century | Iraq | √ | | | | Three face NE, and one faces NW | Flat roof |
| 5 | Institute of Science and Technology at Masdar City | Educational | Modern | United Arab Emirates | | | | √ | All | Flat roof |
| 6 | Khalifa stadium | Sports | Modern | Qatar | | | | √ | All | Flat roof |

√ = Yes

## 5. Discussion and Conclusions

Although building height differences and homogeneity in the urban area are important factors for urban ventilation performance in an area [49], as studied in old cities such as Elazığ, this study did not address this point since urban morphology surrounding the case studies has changed throughout history. Therefore, this study relied on a wind rose analysis for each region.

In conclusion, the wind roses in the regions studied in this paper explain why the wind towers were designed the way they were. Some regions, such as Cairo, Dubai, Baghdad, and Abu Dhabi, receive wind in mostly one primary direction. Therefore, a one-sided or two-sided windcatcher is the prevalent choice in Cairo. Using two-sided windcatchers that are open on both the north and the west sides is especially beneficial because it supplies the building with the wind in the direction that ranges from north to northwest to west, which are the three most prevalent wind directions in Cairo. These results highlight that the vernacular windcatchers in Egypt and the Middle East correspond to the prevailing wind directions and the ventilation needs of the connected spaces. However, in Dubai, the use of four-sided windcatchers contradicts Dubai's wind rose, which shows a limited range in wind direction (even more limited than that of Cairo's). This could be because the Dubai windcatchers were built by Persian immigrants who were probably used to constructing four-sided windcatchers in Iran. Other types with more sides are better in regions with variable wind directions, such as Yazd, Iran [50].

Another aspect guiding the windcatcher design is its relationship to other spaces.

In Egypt, most of the windcatchers open directly into an enclosed space, whereas in other regions of the Middle East, windcatchers are used differently, where most of them are attached to open spaces or courtyards. This may also explain why windcatchers in Egypt are mostly unidirectional since they ventilate one specific space. In other regions, even if there is one direction for prevailing wind, we found multi-directional windcatchers, which ventilate a larger attached open space.

In Egypt, many modern windcatchers followed the design standards of medieval windcatchers. This can be evidence of the success of this design, albeit simple. On the other hand, Hassan Fathy tried to develop the vernacular windcatcher to have its openings face opposite directions. A study shows that an iteration similar to that resulted in the highest air change per hour [50]. Furthermore, the tables above show that 37.5% of the case studies on modern windcatchers have flat roofs. Further studies should be performed to measure the effectiveness of modern flat roof windcatchers in the climate of Egypt.

In the gulf, where the wind is scarce and the climate is harsh, new technologies are added to the wind tower to cool outdoor spaces rather than indoor spaces, such as in the cases of Masdar City in the UAE and Khalifa Stadium in Qatar.

A limitation of this study is that most of the windcatchers from the Pharaonic to the Medieval eras did not survive in Egypt since they were mainly made of wood or reed [12]. However, the cases of both the Madrasa of al-Nāṣir Muḥammad and The Khanqah of Sultan Baybars al-Jashankir, which do not have surviving windcatchers [12,19,24,26] prove their prevalent use during the Medieval era. There is evidence of the use of windcatchers in cases such as those in the presence of wind ducts. An extension for this research may include using extensive field and numerical studies, which could provide more insight into the design of historical windcatchers, thus advancing the research on sustainable vernacular elements such as the windcatcher. Additionally, combining a windcatcher analysis with current research on the effect of urban configurations on airflow and ventilation [49] could help provide new information on how to utilize these elements to improve ventilation in existing and new neighborhoods.

**Author Contributions:** Conceptualization, M.A.N., A.E. and V.B.; methodology, M.A.N., A.E. and V.B.; formal analysis, A.E and M.A.N.; investigation, A.E. and M.A.N.; resources, A.E., V.B., M.A.N. and N.S.; data curation, A.E., V.B. and N.S.; writing—original draft preparation, A.E., N.S., M.A.N. and V.B.; writing—review and editing, V.B., A.E., M.A.N., S.G. and K.T.; visualization, M.A.N. and A.E.; supervision, S.G. and M.A.N.; project administration, A.E., M.A.N. and S.G.; funding acquisition, S.G and K.T. All authors have read and agreed to the published version of the manuscript.

**Funding:** This research was funded through an internal grant from the American University in Cairo offered through the Office of the Dean of the School of Sciences and Engineering (2021–2022). Supplementary funding from the Office of the Associate Provost for Research, Innovation, and Creativity funded the APC.

**Data Availability Statement:** Not applicable.

**Acknowledgments:** The authors would like to thank the Office of the Dean of the School of Sciences and Engineering at the American University in Cairo for the funding offered to this project and the Office of the Associate Provost for Research, Innovation, and Creativity for the additional publication support funds revived. The authors would also like to thank the Rare Books and Special Collections Library at the American University of Cairo for giving us access to materials in their collections that have helped illustrate this manuscript. The authors would like to thank and acknowledge the excellent English editing of Laila El Refai. The authors would like to thank and acknowledge the technical support offered by the collaborating members: Omar AbdelAziz, Khaled Nassar, Moataz ElDakroury, and Ahmed Hafez.

**Conflicts of Interest:** The authors declare no conflict of interest. The funders had no role in the design of the study; in the collection, analyses, or interpretation of data; in the writing of the manuscript, or in the decision to publish the results.

## Glossary

| | |
|---|---|
| *Diwan-Khanqah(s)* | a reception hall plus hostel in medieval Islamic buildings |
| *Durqa'a(s)* | small, covered court in medieval Islamic buildings |
| *Imam(s)* | the person who leads prayers in a mosque |
| *Iwan(s)* | a vaulted portal opening into a courtyard |
| *Khanqah(s)* | a hostel for Sufis in medieval Islamic buildings |
| *Madrasa(s)* | a school in medieval Islamic buildings |
| *Mashrabiya(s)* | a type of projecting window in medieval Islamic buildings with carved wood latticework |
| *Malqaf(s)* | name used for wind towers in Egypt |
| *Maglis(es)* | a siting room for gatherings in medieval Islamic buildings |
| *Mihrab(s)* | a niche in the wall that points towards the direction of Mecca in a mosque, used to point the congregation during prayers |
| *Salsabil(s)* | a water fountain in medieval Islamic buildings |
| *Qa'a(s)* | a hall in medieval Islamic buildings |
| *Zir(s)* | a pot made of pottery for storing cool water in old houses |

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
