# Peer review of "The Rise and Evolution of Wind Tower Designs in Egypt and the Middle East"

_sustainability, doi:10.3390/su151410881_

Round 1
Reviewer 1 Report
The manuscript presents the development of windcatcher designs in Egypt and the Middle East. I found it very interesting. Though the designs were well presented, technical aspects can be improved. I would suggest the following points:
1) Introduction:
- A figure or sketch to show how a windcatcher works, in addition to the text: "As mention...in the process" should be added.
- "...two-sided wind catcher...two main forces: buoyancy and wind": The descriptions in the text in sections 2&3 are mainly for the wind driving force. Discussions for the buoyancy force should be elaborated, particularly for windcatchers with two-sided and multiple openings.
2) Sections 2-3:
- In some figures, the flow directions were presented (figures 4,9...) but in many others, it is difficult to see or guess (figures 3,8,14...). Arrows indicating flow directions should be added when possible.
- Figure 8: is it a plan or section view?
- Figure 9: "Section A-A", but I can not see the A-A in the sketch.
3) Section 4:
Tables 2 and 3 list the studied cases. However, it is difficult to see how the "evolution" (as in the Title) of the design through the list. Additional discussions about whether the modern designs improve the performance and how the latter designs evolve based on the former ones should be elaborated.
4) Section 6: Concluding remarks
- I seems to me that Sections 5 and 6 can be combined, as section 6 continues the discussions.
- Therefore, the conclusions in section 6 should be revised to present the main findings, limitations, and possible extensions of the study.
Author Response
We would like to express our sincere gratitude for your valuable time, as well as for your useful suggestions aimed at improving our work. We found your comments to be very helpful and believe that the revised paper has benefitted a great deal from your input. We have carefully revised the manuscript to address your concerns and recommendations. In our responses below, we explain in detail how we have addressed each of your points. For your convenience, we repeat your comments in standard typescript, followed by our responses in blue fonts.
Reviewer #1:
The manuscript presents the development of windcatcher designs in Egypt and the Middle East. I found it very interesting. Though the designs were well presented, technical aspects can be improved. I would suggest the following points:
- Introduction:
- A figure or sketch to show how a windcatcher works, in addition to the text: “As mention...in the process” should be added.
Thank you very much for your suggestion.
New figures were added explaining this.
- “...two-sided wind catcher...two main forces: buoyancy and wind”: The descriptions in the text in section 2&3 are mainly for the wind driving force. Discussions for the buoyancy force should be elaborated, particularly for windcatchers with two-sided and multiple openings.
Thank you for your comment and for sharing your concern in that regard.
The discussion on buoyancy has been elaborated about what buoyancy is, how it works, and its impact on multiple-sided wind towers.
- Sections 2-3:
- In some figures, the flow directions were presented (figures 4,9...) but in many others, it is not easy to see or guess (figures 3,8,14...). Arrows indicating flow directions should be added when possible.
Thank you very much for your suggestion. Flow directions are added when possible. However, adding flow directions for all wind towers might be misleading since their operation is sometimes affected by wind speed and direction.
- Figure 8: is it a plan or section view?
Thank you for your comment. This is a section. This was clarified in the caption.
- Figure 9: “Section A-A”, but I can not see the A-A in the sketch
Thank you for your comment . This was addressed in the revisions.
- Section 4:
- Tables 2 and 3 list the studied cases. However, it is difficult to see how the “evolution” (as in the Title) of the design through the list. Additional discussions about whether the modern designs improve the performance and how the latter designs evolve based on the former ones should be elaborated
Thank you for your comment and for sharing your concern in that regard.
The discussion about whether the modern designs improve performance and how the latter designs evolve based on the former ones is elaborated upon further in this section and the “Discussion and Conclusion” section.
- Section 6: Concluding remarks
- I seems to me that Sections 5 and 6 can be combined, as section 6 continues the discussions.
- Therefore, the conclusions in section 6 should be revised to present the main findings, limitations, and possible extensions of the study.
Thank you for your comment and suggestions.
We have reviewed the conclusion and reworded the subtitles of these sections accordingly. We also added the study’s main findings, limitations, and possible extensions.
Reviewer 2 Report
Authors performed a study on The rise and evolution of wind tower designs in Egypt and the Middle East.
The study needs following revisions before publish as:
It is better if authors give chosen parameters in abstract.
Subtitles can be removed from Introduction.
Recent works can be added. For example, the study of Analysis of the natural ventilation performance of residential areas considering different urban configurations in Elazig, Turkey can be useful for authors.
Figures in Table 1 are not clear. Please replot them.
Fig. 4 is not clear. There is no dimensions or boundary conditions.
Number of figures can be reduced.
Please check line 573. It is unclear words.
Then, manuscript can be accepted in this journal.
English level is acceptable.
Author Response
We would like to express our sincere gratitude for your valuable time, as well as for your useful suggestions aimed at improving our work. We found your comments to be very helpful and believe that the revised paper has benefitted a great deal from your input. We have carefully revised the manuscript to address your concerns and recommendations. In our responses below, we explain in detail how we have addressed each of your points. For your convenience, we repeat your comments in standard typescript, followed by our responses in blue fonts.
Reviewer #2:
Authors performed a study on The rise and evolution of wind tower designs in Egypt and the Middle East. The study needs following revisions before publish as:
- It is better if authors give chosen parameters in abstract.
Thank you for your comment and suggestion.
The abstract has been modified accordingly, highlighting that the chosen wind tower parameters of investigation in the paper are shapes, number of sides, and orientation.
- Subtitles can be removed from Introduction.
Thank you for your comment and suggestion.
We removed the subtitles, and we completely agree that the introduction reads much better as one text.
- Recent works can be added. For example, the study of Analysis of the natural ventilation performance of residential areas considering different urban configurations in Elazig, Turkey can be useful for authors.
Thank you for your comment and for sharing your concern in that regard.
- Figures in Table 1 are not precise. Please replot them.
Thank you for your comment and for sharing your concern in that regard.
We have replaced the figures with clearer ones.
- Fig. 4 is not clear. There is no dimensions or boundary conditions.
Thank you for your comment and for sharing your concern in that regard.
We have made sure that a graphical scale appears in the figure. We hope this addresses your concern.
- Number of figures can be reduced.
Thank you for your comment and for sharing your concern in that regard.
We have reduced the number of figures to the necessary ones.
- Please check line 573. It is unclear words.
Thank you for your comment and for sharing your concern in that regard.
We have revised this part.
- Then, manuscript can be accepted in this journal.
Thank you for recommending the paper for publication.
Reviewer 3 Report
Dear Authors,
I appreciate the opportunity to review your research paper titled "The Rise and Evolution of Wind Tower Designs in Egypt and the Middle East: Implications for Sustainability in Heritage and Urban Planning." Your study delves into an intriguing subject and presents valuable insights into the historical development of wind towers. While the paper exhibits several strengths, I would like to suggest a few aspects that, if addressed, could further enhance its overall quality and impact.
1. First and foremost, I recommend that you enhance the introduction section by explicitly addressing certain key aspects. It would be beneficial to clarify the necessity of your study and why it is relevant in the current academic and practical landscape. By explicitly stating the importance of your research, you will effectively establish its significance and the reasons behind its execution.
a. Furthermore, it would be valuable to highlight the main literature gaps that you have identified, which have motivated you to undertake this study. By discussing these gaps, you will provide readers with a better understanding of the existing knowledge and the unique contributions your research makes in filling those gaps.
b. In addition, I encourage you to clearly outline the research's originality and novelty. What sets your study apart from previous works in the field? How does it contribute to advancing the knowledge and understanding of wind tower designs? Articulating the unique aspects of your research will help readers appreciate its originality and innovative nature.
c. To provide readers with a clear roadmap of your paper, I suggest including a paragraph that outlines the structure of the paper. This will guide readers through the subsequent sections and ensure a smooth flow of information. It will also help readers navigate the content more effectively and comprehend the logical progression of your arguments.
By incorporating these suggestions into your introduction, you will strengthen the foundation of your research and provide readers with a comprehensive understanding of its necessity, originality, and structure. This, in turn, will enhance the overall readability and impact of your paper.
2. In the Results section, it is important to provide clear and concise descriptions of the outcomes of your study. Address the following points to provide a comprehensive understanding of the results:
a. Clearly state what transpired during the study.
b. Discuss the discoveries or confirmations made through the analysis.
c. Present the data in simple terms, emphasizing the main observations.
d. Include any statistical analyses conducted, such as significance or goodness-of-fit measures.
e. Evaluate the observed trends and explain the significance of the results in the context of broader understanding, referring to relevant published research.
f. Offer a critical analysis of the collected data, acknowledging any limitations or potential biases.
3. In the Discussions section, engage in a comprehensive analysis of the results and provide insights that contribute to the existing knowledge in the field. Consider the following suggestions:
a. Identify any gaps or inconsistencies in the research and address them appropriately.
b. Suggest ways in which future research can confirm the conclusions or further advance the study.
c. Relate your findings to existing literature and theories, discussing their implications and potential contributions to both theoretical and practical aspects.
d. Encourage critical thinking and explore possible alternative explanations or interpretations of the results.
e. Synthesize the results and the implications they have for the broader field of wind tower designs and sustainable construction.
4. The Conclusions section should effectively summarize the key takeaways of your research. Consider the following elements when formulating this section:
a. Ensure that the conclusions align with the objectives of your study, regardless of whether they were fully achieved or not.
b. Highlight the contributions made by your research to the field of wind tower designs and sustainable construction.
c. Discuss the theoretical and practical implications of your findings.
d. Provide directions for future research, identifying potential areas of exploration and unanswered questions.
5. By incorporating more recent and relevant references, you will strengthen the scientific foundation of your research paper and demonstrate your engagement with the current literature. This will contribute to the overall quality and impact of your study.
By incorporating these suggestions, you will strengthen the structure of your paper and provide readers with a cohesive and impactful research narrative.
Once again, I commend you on your work and appreciate the opportunity to provide feedback. I believe that by addressing these aspects, your research paper will reach an even higher level of excellence.
Minor editing of English language required
Author Response
We would like to express our sincere gratitude for your valuable time, as well as for your useful suggestions aimed at improving our work. We found your comments to be very helpful and believe that the revised paper has benefitted a great deal from your input. We have carefully revised the manuscript to address your concerns and recommendations. In our responses below, we explain in detail how we have addressed each of your points. For your convenience, we repeat your comments in standard typescript, followed by our responses in blue fonts.
Reviewer #3:
I appreciate the opportunity to review your research paper titled “The Rise and Evolution of Wind Tower Designs in Egypt and the Middle East: Implications for Sustainability in Heritage and Urban Planning.”
Your study delves into an intriguing subject and presents valuable insights into the historical development of wind towers. While the paper exhibits several strengths, I would like to suggest a few aspects that, if addressed, could further enhance its overall quality and impact.
- First and foremost, I recommend that you enhance the introduction section by explicitly addressing certain key aspects. It would be beneficial to clarify the necessity of your study and why it is relevant in the current academic and practical landscape. By explicitly stating the importance of your research, you will effectively establish its significance and the reasons behind its execution.
- Furthermore, it would be valuable to highlight the main literature gaps that you have identified, which have motivated you to undertake this study. By discussing these gaps, you will provide readers with a better understanding of the existing knowledge and the unique contributions your research makes in filling those gaps.
- In addition, I encourage you to clearly outline the research’s originality and novelty. What sets your study apart from previous works in the field? How does it contribute to advancing the knowledge and understanding of wind tower designs? Articulating the unique aspects of your research will help readers appreciate its originality and innovative nature.
- To provide readers with a clear roadmap of your paper, I suggest including a paragraph that outlines the structure of the paper. This will guide readers through the subsequent sections and ensure a smooth flow of information. It will also help readers navigate the content more effectively and comprehend the logical progression of your arguments.
- By incorporating these suggestions into your introduction, you will strengthen the foundation of your research and provide readers with a comprehensive understanding of its necessity, originality, and structure. This, in turn, will enhance the overall readability and impact of your paper.
- In the Results section, it is important to provide clear and concise descriptions of the outcomes of your study. Address the following points to provide a comprehensive understanding of the results:
- Clearly state what transpired during the study.
- Discuss the discoveries or confirmations made through the analysis.
- Present the data in simple terms, emphasizing the main observations.
- Include any statistical analyses conducted, such as significance or goodness-of-fit measures.
- Evaluate the observed trends and explain the significance of the results in the context of broader understanding, referring to relevant published research.
- Offer a critical analysis of the collected data, acknowledging any limitations or potential biases.
- In the Discussions section, engage in a comprehensive analysis of the results and provide insights that contribute to the existing knowledge in the field. Consider the following suggestions:
- Identify any gaps or inconsistencies in the research and address them appropriately.
- Suggest ways in which future research can confirm the conclusions or further advance the study.
- Relate your findings to existing literature and theories, discussing their implications and potential contributions to both theoretical and practical aspects.
- Encourage critical thinking and explore possible alternative explanations or interpretations of the results.
- Synthesize the results and the implications they have for the broader field of wind tower designs and sustainable construction.
- The Conclusions section should effectively summarize the key takeaways of your research. Consider the following elements when formulating this section:
- Ensure that the conclusions align with the objectives of your study, regardless of whether they were fully achieved or not.
- Highlight the contributions made by your research to the field of wind tower designs and sustainable construction.
- Discuss the theoretical and practical implications of your findings.
- Provide directions for future research, identifying potential areas of exploration and unanswered questions.
- By incorporating more recent and relevant references, you will strengthen the scientific foundation of your research paper and demonstrate your engagement with the current literature. This will contribute to the overall quality and impact of your study.
By incorporating these suggestions, you will strengthen the structure of your paper and provide readers with a cohesive and impactful research narrative.
Once again, I commend you on your work and appreciate the opportunity to provide feedback. I believe that by addressing these aspects, your research paper will reach an even higher level of excellence.
Thank you for your detailed feedback on our manuscript. We have taken the time to read your suggestions and comments and have considered them and made the following changes.
- Introduction: We have added significant writing to the introduction in order to provide more coverage for the current academic knowledge on the topic and have clarified further the scope of the work, as well as adding a paragraph that outlines the structure of the paper.
- Results: we have restructured the manuscript to highlight further the results (which are an outcome of our case study analysis) and explained the observed trends. We also added some text to contextualize our findings further.
- Discussion and conclusion: we have merged those two sections to provide an overview of the main findings and the limitations of the work. We also contextualized the findings within the existing literature, suggesting areas of future research. The combined section also now highlights the implications our work has on the broader field of wind tower designs and sustainable construction.
- References: We have incorporated more recent and relevant references to strengthen the research and to demonstrate its engagement with recent literature.
We hope that these modifications have allowed us to address your comments. We thank you again for your feedback.
Round 2
Reviewer 1 Report
Dear authors,
Thank you very much for very kind responses to my questions and comments. I found the manuscript much improved and it can be considered for publication with this journal.
Best regards,